# Exploring amino acid functions in a deep mutational landscape

Alistair S Dunham ⬤ & Pedro Beltrao* ⬤

## Abstract

Amino acids fulfil a diverse range of roles in proteins, each utilising its chemical properties in different ways in different contexts to create required functions. For example, cysteines form disulphide or hydrogen bonds in different circumstances and charged amino acids do not always make use of their charge. The repertoire of amino acid functions and the frequency at which they occur in proteins remains understudied. Measuring large numbers of mutational consequences, which can elucidate the role an amino acid plays, was prohibitively time-consuming until recent developments in deep mutational scanning. In this study, we gathered data from 28 deep mutational scanning studies, covering 6,291 positions in 30 proteins, and used the consequences of mutation at each position to define a mutational landscape. We demonstrated rich relationships between this landscape and biophysical or evolutionary properties. Finally, we identified 100 functional amino acid subtypes with a data-driven clustering analysis and studied their features, including their frequencies and chemical properties such as tolerating polarity, hydrophobicity or being intolerant of charge or specific amino acids. The mutational landscape and amino acid subtypes provide a foundational catalogue of amino acid functional diversity, which will be refined as the number of studied protein positions increases.

**Keywords** amino acid function; deep mutational scanning; mutational consequences; protein structure; unsupervised learning
**Subject Categories** Computational Biology; Methods & Resources
**Mol Syst Biol. (2021) 17: e10305**

## Introduction

Amino acids fulfil a diverse range of roles in proteins, each utilising its chemical properties in different ways in different contexts in order to create required protein functions. Classically, amino acids split into five broad groups based on their physicochemistry: aliphatic, aromatic, polar, positively charged and negatively charged. In addition, proline and glycine uniquely form a backbone loop and lack a side chain, respectively. However, the same amino acid can have different effects in different contexts. For example, consider three tyrosine contexts: Tyr197 in the active site of NADPH dehydrogenase, a tyrosine phosphosite and Tyr198 in inositol polyphosphate multikinase. The first two use the hydroxyl group in divergent ways, forming a stabilising hydrogen bond to the enzymes substrate and covalently bonding to the new phosphate group during phosphorylation. The final example uses a completely different property, the aromatic ring, to form a stabilising interaction with a sulphur atom in a nearby methionine (Weber & Warren, 2019). Even in the narrow context of catalytic sites, the same amino acids have been shown to perform a range of active roles across the proteome (Ribeiro et al, 2020). The role an amino acid fills at a given position is defined by its chemical environment, which is determined by many factors including nearby amino acids, secondary structure, post-translational modifications and bound ligands. Taking account of this environment helps predict whether a site is functional, thus demonstrating the environment's importance (Rice & Eisenberg, 1997; Torng & Altman, 2019).

An amino acid's role affects the consequences of different substitutions at that position, known as the position's mutational profile. Evolutionary conservation data confirms this, with models of substitution likelihood that account for protein context performing better than universal models (Rice & Eisenberg, 1997; Müller et al, 2001; Huang & Bystroff, 2006). Similarly, on a gene-by-gene level, alanine scanning experiments have shown the power of mutations to infer positional functions in proteins using mutations, for example mapping hGH receptor interactions (Cunningham & Wells, 1989) and the CD4-binding site (Ashkenazi et al, 1990). This association between mutational consequences and an amino acid's contextual properties is biologically important, both for understanding protein chemistry and in predicting the consequences of mutations.

Given the strong link between a position's structural and functional role and mutational consequences, the landscape of mutational profiles across the proteome can be used for an unbiased, proteome-wide exploration of the functional diversity of amino acid roles. This analysis would give us an insight into the range and frequency of biochemical roles beyond what can be achieved by analysing individual proteins. Until recently, experimentally characterising this landscape was difficult, but the new deep mutational scanning technique (Fowler & Fields, 2014) measures mutational profiles at very high throughput. These experiments directly measure fitness for all possible variants in a protein or region by

European Molecular Biology Laboratory, European Bioinformatics Institute (EMBL-EBI), Cambridge, UK
*Corresponding author. Tel: +44 1223 494 610; E-mail: pbeltrao@ebi.ac.uk

generating a comprehensive mutant library and subjecting it to selection where survival is dependent on the target protein's function. Sequencing the library before and after selection determines the fitness of variants by measuring whether they have been enriched or depleted relative to the wild type during selection. Data from these experiments have been used to explore properties of the genes being studied, to determine general properties of insertion types (Gray *et al*, 2017) and for training variant effect predictors (Gray *et al*, 2018).

Here, we explore the combined mutational landscape of 30 genes using 33 deep mutational scans from 28 studies. We show that multiple datasets can be meaningfully combined and use them to explore the diverse functions of each amino acid. First, suitable studies were selected and their scores normalised to a common scale. Second, we derived and analysed their combined mutational landscape, relating it to a range of biophysical properties. Finally, we explored the diversity of amino acid roles using the mutational landscape, clustering each amino acid into typical subtypes and demonstrating these subtypes represent meaningful biological groups.

## Results

### A unified deep mutational scanning landscape

The first step to utilising a dataset drawn from many deep mutational scanning studies is combining them in a meaningful way, allowing scores to be compared. We selected 33 deep mutational scans from 28 studies covering 6,291 positions in 30 proteins across nine species (Fig 1A, Dataset EV1). These proteins have a broad range of structures and functions, meaning the analysis applies to general protein properties rather than those of a single protein or protein family. We required that studies applied selection pressures directly related to natural function and that their fitness scores could be transformed to a common scale. To prepare each study (Fig 1B), we first averaged replicates. When multiple selection pressures were measured, we either averaged comparable conditions or preferred those selecting for broader aspects of protein function. For example, when processing data on BRCA1 from Starita *et al* (2015), we used measurements of overall E3 ubiquitin ligase activity rather than BARD1 binding because it may reflect protein function more generally. Conditions were considered comparable when they test similar aspects of protein function and have correlated results, for example different drug types in Melnikov *et al*, (2014). Some studies generated sequences with multiple variants, so when individual

substitutions were not explicitly measured, we averaged across variant sequences containing each substitution, after validating that this produces approximately correct scores for explicitly measured variants (Appendix Fig S1). Next, substitution fitness scores were transformed into a common enrichment ratio (ER) score, which measures the enrichment of a variant during selection relative to the change in the wild type. Thus, positive ER scores mean the variant is advantageous, zero is neutral and negative scores deleterious. We normalised scores from each study against the median of the lowest 10% of scores, reasoning that the worst substitutions (e.g. nonsense mutations) result in complete loss of function and are comparable between studies. We excluded 626 positions (9% of the starting dataset) with scores for < 15 of the possible 20 nonsynonymous substitutions (including nonsense) to focus on positions with sufficient data and then imputed the remaining missing data (see Methods). This resulted in complete sets of normalised ER scores for substitutions to all 20 amino acids at 6291 unique positions across 30 genes, with 98.16% of nonsynonymous ER scores experimentally measured and the remaining 1.84% imputed. We refer to the vector of a position's ER scores as its mutational profile and the combined mutational profiles for all positions and proteins as the mutational landscape.

Three genes were covered by multiple studies (Fig 1C and Appendix Fig S2). These were HSP90 (a protein chaperone), TEM1 (a GTPase involved in mitosis) and ubiquitin (UBI), which is added to proteins by ubiquitin ligase enzymes as a function altering post-translational modification. The scores from these results were sufficiently correlated (HSP90 $r^2 = 0.4038$, TEM1 $r^2 = 0.994$, UBI $r^2 = 0.4676$) to suggest scores are robust and represent variants' biological properties. Some of the differences between these are likely due to the selection criteria. For example, one study measured UBI fitness via surface display and E1 ubiquitin ligase activity (Roscoe & Bolon, 2014) and the other used growth in the absence of WT UBI (Roscoe *et al*, 2013). The two TEM1 studies were so similar to each other that only the most recent was used. On the other hand, the HSP90 and UBI studies were sufficiently different that they could all be retained and used to check consistency of later results. The fact that scans under different selection criteria and experimental conditions correlate suggests many protein properties are somewhat independent of conditions, for instance relating instead to the basic structure of the protein.

Deep mutational scanning results should relate to evolutionary conservation, as natural selection also samples variants and enriches them based on fitness. We validated our approach to combine DMS datasets by showing this relationship was maintained in bulk statistics and for individual variants in our data. Substitution matrices,

---

**Figure 1. Combining deep mutational scanning studies.**

A  Proteins in the combined dataset, with the structure used, number of positions with mutational profiles and the percentage of these in the structure model.

B  Normalisation pipeline.

C  Correlation between ER scores from two deep mutational scans on ubiquitin, both from the Bolon Lab ($r^2 = 0.4676$, Pearson's correlation coefficient *t*-test: $P < 2.2 \times 10^{-16}$). A simple linear regression best fit is shown, with a 95% confidence interval in grey shading.

D  Relationship between mean normalised ER score for an amino acid substitution and corresponding BLOSUM62 score. The scores for missense variants strongly correlate ($r^2 = 0.4194$, Pearson's correlation coefficient *t*-test: $P < 2.2 \times 10^{-16}$). A simple linear regression best fit is shown, with a 95% confidence interval in grey shading.

E  Correlation between normalised ER score and $\log_{10}$SIFT4G score in each study based on Pearson correlation coefficient. The number of variants considered for each gene is indicated above each column. Error bars indicate the confidence interval for Pearson's ρ based on Fisher's *Z* transform. *P*-values are calculated using Pearson's correlation coefficient *t*-tests.

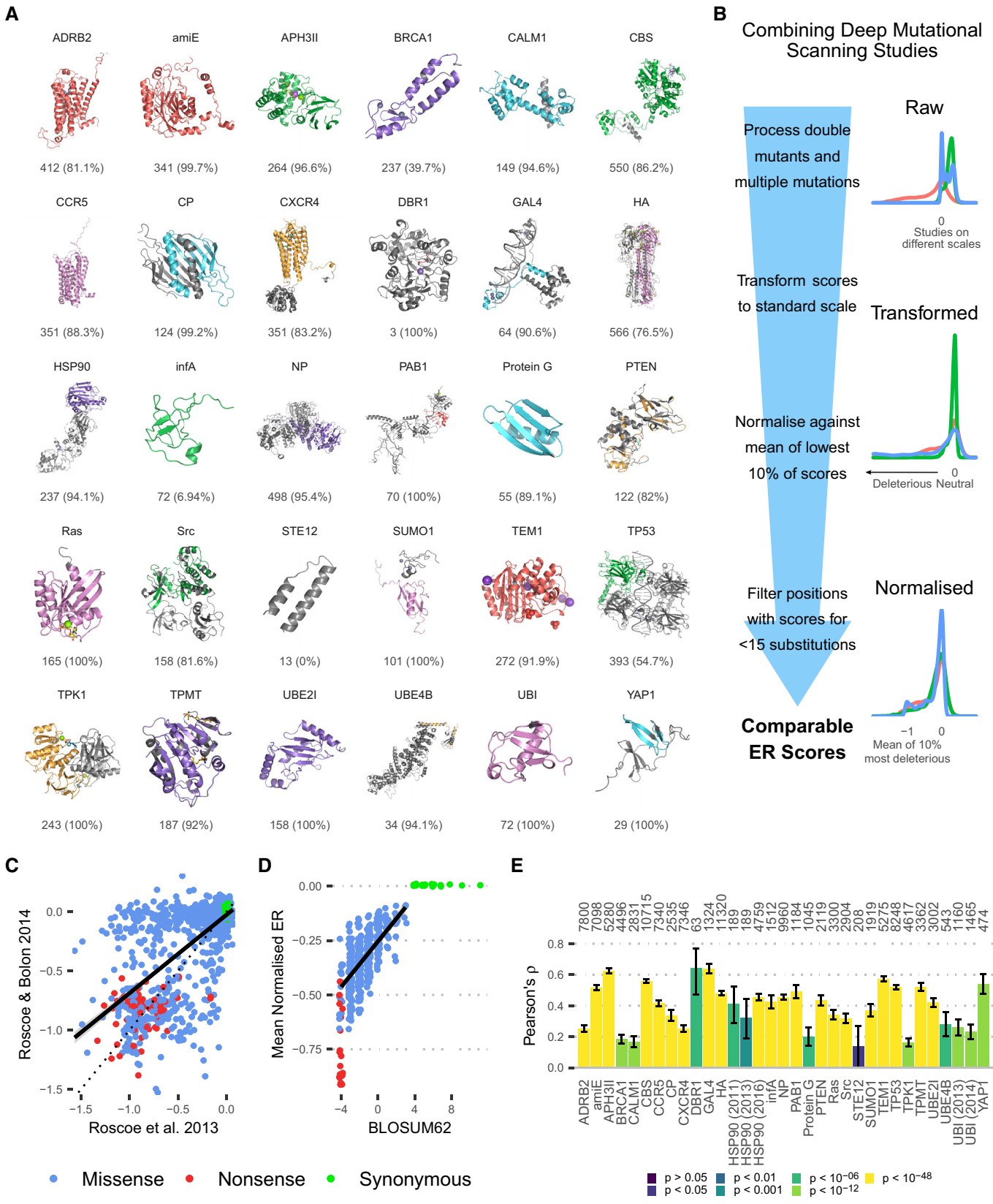

**Figure 1.**

such as BLOSUM62 (Henikoff & Henikoff, 1992), measure how frequently each amino acid is mutated to any other, aggregated across a large number of sequence alignments. The mean ER score for each missense substitution type (A → C, A → D, . . .) in our dataset correlates well with BLOSUM62 (Fig 1D, $r^2 = 0.4194$, $P < 2.2 \times 10^{-16}$). BLOSUM62 also provides substitution scores for synonymous substitutions, which measure the average conservation of positions of each amino acid. While synonymous substitutions tend to have close to 0 ER in deep mutational scans (92% have |ER| < 0.05 in this dataset), it is possible to estimate the impact of mutating away from each amino acid by calculating the mean ER score for all substitutions away from that amino acid. This score correlates with synonymous BLOSUM62 scores ($r^2 = 0.3191$, $P = 0.009452$). Secondly, we used SIFT4G (Vaser et al, 2015) to predict the impact of specific substitutions at each position. SIFT4G uses an alignment of homologous sequences to predict the deleterious impact of a substitution based on how often the alternate amino acid is observed at the mutated position in the alignment. It outputs a probability score ranging from 0 to 1 with scores below 0.05 suggesting the variant is likely to be deleterious. As expected, the experimentally determined ER scores for specific substitutions correlate with the predicted impact based on the $\log_{10}$SIFT4G scores (Fig 1E). The varying correlation with SIFT4G scores between studies suggests that the different experimental selection pressures vary in the degree they mirror natural evolutionary pressures.

The combined dataset gives us a large number of mutational profiles, together constituting a subset of the overall proteome mutational landscape. While it covers a relatively small subset of the vast proteome space, the dataset does cover a reasonably varied and representative sample, with proteins covering a range of species, functions and environments. Thus, we generated an approximation for the mutational landscape of proteins that can be used to analyse the range and frequency of amino acid roles in an unbiased manner.

## Biophysical relationships in the deep mutational landscape

The link between mutational consequences and amino acid's roles means the mutational landscape we have derived is a quantitative representation of the diverse roles amino acids can play in proteins. Clustering all protein positions by their mutational outcomes is expected to identify positions across different proteins that have similar mutational properties and therefore likely share similar biophysical properties. The relationships between protein positions can be visualised by dimensionality reduction of the mutational landscape using UMAP (preprint: McInnes et al, 2018) and PCA (Fig 2). In this representation, two protein positions are closer together when they have similar mutational profiles. If the differences in mutational outcomes were strongly determined by experimental approach, positions would group by protein or study of origin. Instead, all studies are distributed across UMAP space, suggesting there is not a strong study bias in the normalised mutational profiles (Appendix Fig S3). In contrast, the same protein positions assayed in different studies are significantly more similar and closer together in UMAP space, on average, than random pairs of positions (Appendix Fig S4, mean 2.105 units closer, one-tailed Mann–Whitney U-test: $P = 1.141 \times 10^{-13}$), which suggests the method is replicable and validates our approach to building a combined dataset.

The relationship between the mutational landscape and the biophysical properties of protein positions is most clearly illustrated by highlighting protein domains in UMAP space (Fig 2A) and observing separation by function. In this case, we highlight positions in three transmembrane proteins (ADRB2, CCR5 and CXCR4) and show they divide primarily by the solvent environment they are exposed to either lipid membrane or hydrophilic intra/extracellular solvent. Quantitative properties can be also demonstrated in the landscape, the strongest of which is the link to mean normalised ER and its link to overall evolutionary conservation at a position. The first UMAP dimension strongly correlates with mean normalised ER ($r^2 = 0.9036$, $P < 2.2 \times 10^{-16}$) and more weakly with mean $\log_{10}$SIFT4G score ($r^2 = 0.2169$, $P < 2.2 \times 10^{-16}$, Fig 2B), which is a measure for evolutionary conservation. Indeed the first principal component (46.7% of the variance, Appendix Fig S5) essentially is mean normalised ER ($r^2 = 0.9991$, $P < 2.2 \times 10^{-16}$).

We computationally derived each position's physical properties from its sequence and structure (Dataset EV2) and demonstrated strong patterns in UMAP space. Positions segregate on the mean hydrophobicity of their wild-type amino acid (Bandyopadhyay & Mehler, 2008) (Fig 2C) and show a strongly related pattern in their surface accessibility (Hubbard & Thornton, 1993) (Fig 2D). FoldX (Schymkowitz et al, 2005) uses a physics derived force field to estimate $\Delta\Delta G$ changes in folding energy after mutation, breaking the result down into components from different physical sources (Van der Waals, electrostatics, etc). The average $\Delta\Delta G$ from each component across all substitutions at a position is indicative of the physical effects of the wild-type amino acid, but with the opposite $\Delta\Delta G$ sign because substitutions disrupt wild-type interactions. For example, positive hydrogen bond $\Delta\Delta G$ values for most substitutions suggest that the wild type makes stabilising hydrogen bonds that are disrupted by mutations. These measurements of positions' physical properties also display strong patterns in UMAP space (Fig 2E and F).

We show that the mutational landscape has rich and complex relationships with a range of biophysical properties and expect there to be others not highlighted in our analysis. Therefore, a position's location in this landscape indicates the likely properties it has, creating a quantitative map of the diverse functions of amino acids in proteins. This makes the landscape a useful tool for analysing new data, identifying likely positional and protein properties and highlighting outlying positions with unusual mutational consequences. Since positions with similar roles or in similar environments group together in the landscape, as seen in domains of ADRB2, CCR5 and CXCR4, reduced dimensionality representations of the mutational landscape are a good space to evaluate the diversity of amino acid roles. It is noteworthy that creating this combined landscape across a range of protein chemistry would not be possible without combining multiple studies.

## Mapping the diversity of amino acid subtypes

Next, we used the mutational landscape to study the diverse roles played by each amino acid, utilising the broad range of proteins in our dataset to cover as much biochemistry as possible. We have shown that positions with similar properties, and therefore likely similar roles, cluster in this space so an approach to mapping role diversity would be to split positions of each amino acid into typical

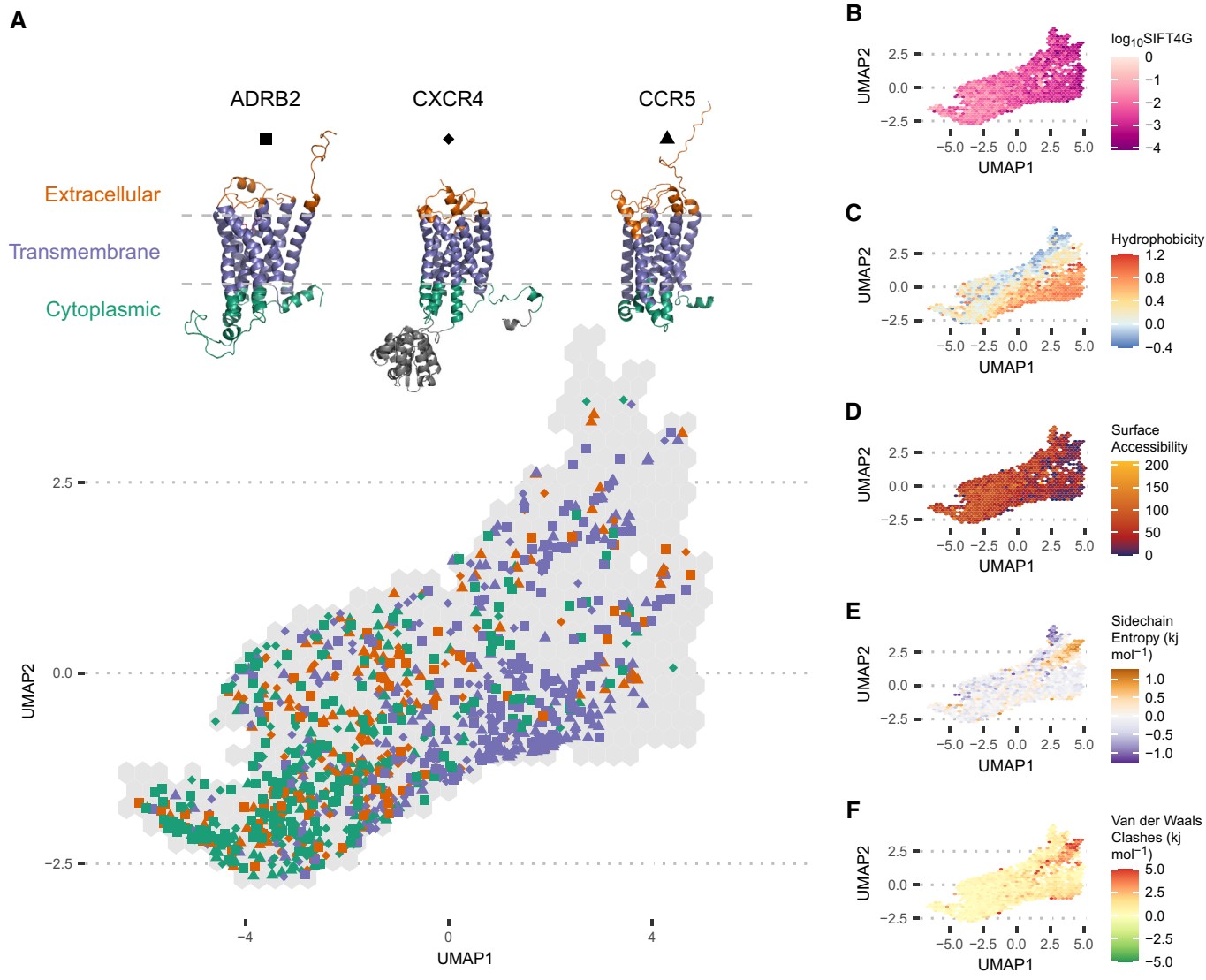

**Figure 2. UMAP projection of the combined deep mutational landscape, coloured by the average of different biophysical factors of protein positions in each hexagonal bin.**

A  Separation of sites in the extracellular, transmembrane and cytoplasmic domains of three transmembrane proteins (ADRB2, CXCR4 and CCR5).

B  Mean $\log_{10}$SIFT4G score, showing how SIFT4G predictions become increasingly deleterious along the first UMAP component ($r^2 = 0.2169$, Pearson's correlation coefficient $t$-test: $P < 2.2 \times 10^{-16}$).

C  Mean amino acid hydrophobicity.

D  All atom absolute residue surface accessibility.

E  Mean sidechain entropy term from all FoldX substitution $\Delta\Delta G$ predictions at a position.

F  Mean Van der Waals clashes term from all FoldX substitution $\Delta\Delta G$ predictions at a position.

subtypes. This would give an overview of typical functions played by each amino acid, their relationships to other amino acids and how frequent they are. We clustered the mutational profiles of each amino acid's wild-type positions independently (Fig 3A, Materials and Methods), in order to study each amino acid's roles in an unbiased manner.

Clustering mutational profiles was dominated by the position's mean ER score, which prevented separation by other properties. To avoid this, we clustered using the principal component representation of mutational profiles, excluding PC1 because it correlates

strongly with mean ER. Permissive positions have low magnitude ER scores for all substitutions and so the balance of other principal components is noisy. To account for this, we split positions where the magnitude of all ER scores is < 0.4 into a permissive subtype for each amino acid. Finally, we use hierarchical clustering and hybrid dynamic tree cutting (Langfelder *et al*, 2008) to identify subtypes (Dataset EV3), which we label XP, XO, X1, X2, ... for permissive positions, outliers and the main subtypes (in frequency order) of amino acid X. This means, for example, that the subtypes labelled 1 are the most frequent for each amino acid, as opposed to sharing

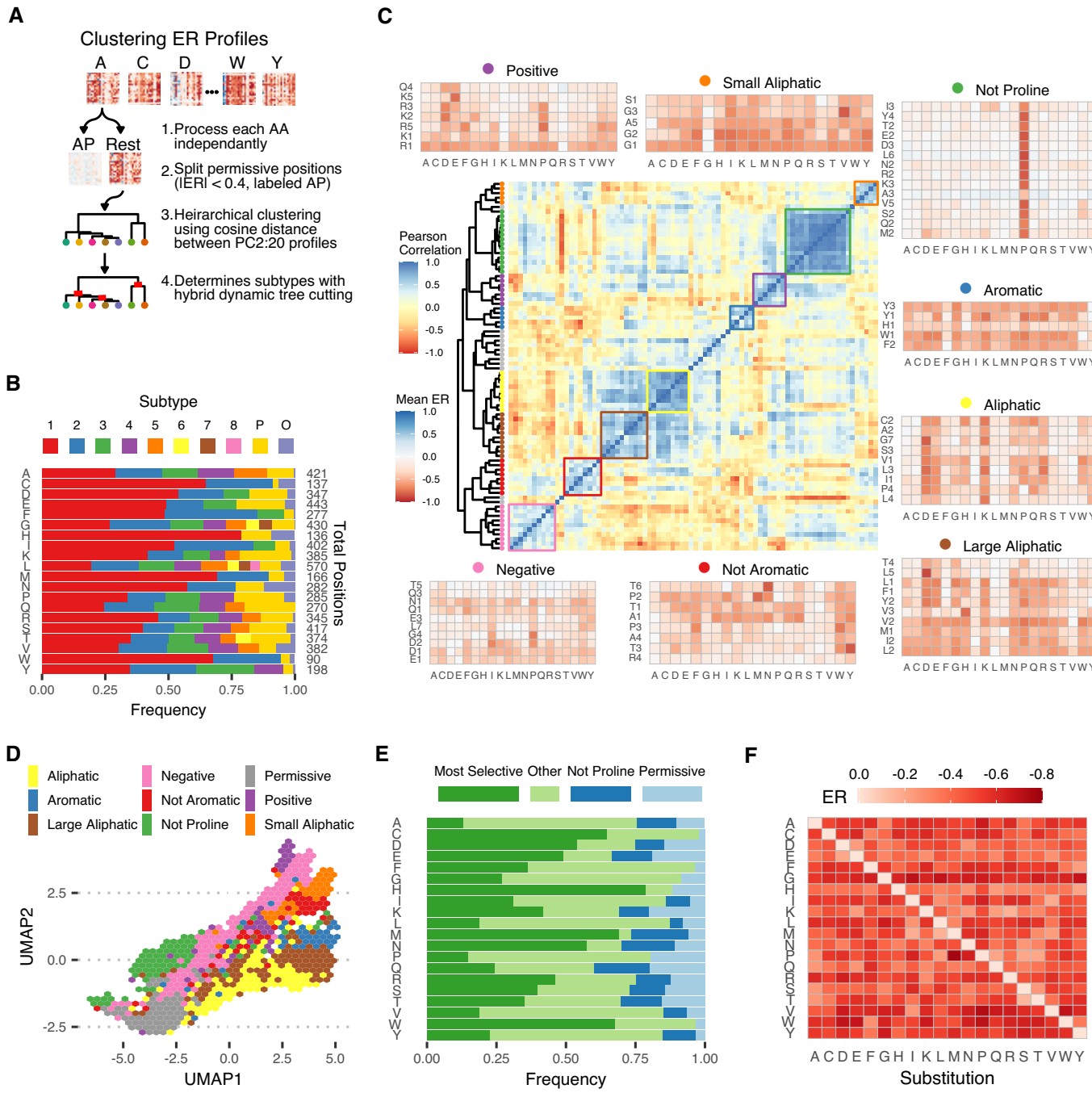

**Figure 3.  Amino acid subtypes.**

A Subtype clustering pipeline.

B Frequency of each subtype and total number of positions of each amino acid. Subtypes are numbered from most to least frequent within each amino acid, with P and O representing permissive and outlier positions.

C The central heatmap shows the clustering of the Pearson correlation between subtypes' mean mutation profiles. The surrounding heatmaps show the mutational profiles (mean ER scores) of subtypes in each group highlighted on the correlation heatmap, as marked with the matching colour.

D Positions from each correlation cluster mapped to the UMAP projection of the deep mutational landscape.

E Frequency of the most selective (highest mean normalised ER: A5, C1, D1, E1, F2, G1, H1, I2, K1, L2, M1, N1, P2, Q1, R1, S1, T1, V2, W1, Y3), not proline (see (C)) and permissive subtypes (labelled P) of each amino acid.

F Mutational profile of the most selective subtype (see (E)) of each amino acid.

functional roles. This approach gave us subtypes with distinct mean mutational profile patterns and therefore different properties. We find between 1 and 8 subtypes for each amino acid, along with permissive positions for all 20 and outlier positions not consistent with any subtype for all but glutamine (Fig 3B). In total, this defines 100 amino acid subtypes, including permissive positions for each amino acid. Histidine is the one amino acid where we only found a single subtype, suggesting it tends to fulfil a similar role in most positions.

We tested the robustness of identification of amino acid subtypes, finding 31 of 66 positions covered by multiple studies were assigned to the same subtype and many others were in two functionally similar subtypes, despite differences in experiments (Appendix Fig S6). This result is extremely unlikely by chance (one-tailed binomial test versus random assignment: $P = 3.397 \times 10^{-4}$), which suggests the procedure is robust. However, some subtype differences might correspond to differences in properties revealed by changes in experimental conditions or selection pressure. To ask how complete is this set of 100 amino acid subtypes, we first observed that the frequency of each successive subtype (Appendix Fig S7A) decreases exponentially across all amino acids ($f = e^{-0.56197 \pm 0.01655 \times N}$, $r^2 = 0.9359$, $P < 2.2 \times 10^{-16}$), suggesting that increasing the size of the dataset would primarily identify less common subtypes. Reclustering the dataset with an increasing number of positions (Appendix Fig S7B) suggests we are not close to identifying all of the subtypes. While the number of subtypes of most amino acids steadily increases as more positions are added that of cysteine and histidine plateaus and tryptophan only increases very slowly (Appendix Fig S7C), suggesting additional subtypes for those amino acids may be less common. Altogether this suggests that a larger dataset would very likely lead to discovery of several more subtypes that are probably rarer than those observed here. Conversely, it is relevant to note that the identification of any meaningful number of subtypes would not be possible without performing the combined analysis of several proteins.

To allow for the future expansion of this study, we have developed the DeepScanScape R package (github.com/allydunham/deepscanscape) to apply the data processing, deep landscape analysis and subtype assignment to aid analysis of future deep mutational scans. This tool also allows for the mapping of positions from a new study to this reference landscape allowing for the predictions of some structural properties.

## Properties and frequency of amino acid subtypes

The DMS datasets combined here allow us to identify and determine the frequency of amino acid subtypes. We then clustered the average mutational profiles of the subtypes (Fig 3C) finding groups with similar mean mutational profiles. These groups of subtypes represent cases where positions from different amino acids have similar mutational properties. For example, they may generally tolerate or select against a specific type of amino acid, for example requiring small aliphatic amino acids or tolerating everything apart from proline (referred to as the small aliphatic group and not proline group, respectively). Subtype groups occupy different regions of UMAP space, thus tending to have different physical properties as well as different mutational profiles (Fig 3D). The most striking example is the "not proline" group, whose positions' only role

appears to be not restricting backbone conformation. These subtypes have a very consistent profile and are found in 14 of the 19 non-proline amino acids, occurring in up to 20.5% of amino acid positions. Methionine (20.5%) and glutamine (20%) have the most "not proline" positions and larger amino acids, such as aromatics (mean 2.9%) or leucine (4.6%) tend to have fewer (Fig 3E). They are often observed to occur in and around secondary structural elements, in particular having a significantly higher likelihood of occurring in alpha helices (mean 1.24 times as likely, one-tailed Mann–Whitney U-test: $P = 1.7 \times 10^{-5}$), based on secondary structure predictions by Porter5 (preprint: Torrisi *et al*, 2018). Another notable group of subtypes only tolerate small aliphatic amino acids. Cross-referencing their positions in UMAP space with our property map (Fig 2) reveals these positions tend to be highly conserved, buried, moderately hydrophobic and lead to Van der Waals clashes when mutated, which is exactly what would be expected of positions that specifically require small amino acids. The other subtype groups identified follow similar patterns, tolerating certain specific groups of amino acids (aliphatic, negative, not aromatic etc.). These subtype groups capture the major divisions of classic amino acid chemistry but, importantly, not all subtypes of each amino acid fall into the group matching their classical chemistry. For example, Y1 and Y3 are in the aromatic subtype group but Y2 tolerates hydrophobicity more broadly, and R1 selects for positive charge but R2 positions primarily select against negative charge rather than requiring the positive charge itself.

The frequencies of the most selective subtype, with the highest mean ER score, and the permissive subtype, defined to be positions with all $|\text{ER}| < 0.4$, vary widely between amino acids (Fig 3E), indicating how often each amino acid fills either highly specific or very general roles. For example, the most selective subtype of cysteine, methionine and tryptophan all occur in at least 65% of their positions in our data, meaning these amino acids are very often used for specific functions. Conversely, a much smaller proportion of glycine positions are the most selective subtype (27%) but that subtypes' mean ER profile (Fig 3F) shows those positions are some of the most selective in the dataset; it is not common for a glycine to be highly selected but when they are they fulfil a very specific role. The frequency of permissive positions varies between amino acids as well. Hydrophobic amino acid positions (e.g. aliphatic and aromatic) tend to be permissive significantly less frequently than other amino acids (mean 0.42 times as likely, one-tailed *t*-test: $P = 0.0001264$, Cohen's $d = 1.37$), with this effect being particularly strong for aromatic amino acids (all < 4% frequency). This may be because they tend to occur in positions that at least require hydrophobicity, such as the core of the protein, and therefore strongly hydrophilic substitutions are selected against. Permissive subtype positions also tend to be more surface accessible in general (mean 28.9Å$^2$ greater accessible surface area, one-tailed *t*-test $P < 2.2 \times 10^{-16}$, Cohen's $d = 0.61$). The fact that hydrophobic amino acids are often experimentally tolerated on the surface but selected against in nature suggests the experiments may be missing some natural selection pressures.

## Characterising amino acid subtypes

The results so far show that broad subtype analysis can shed light on general trends in amino acid roles, but more targeted analysis is

required to understand subtypes' specific properties. We have chosen three groups of subtypes to cover in detail. Firstly, cysteine positions are divided between two main subtypes. The larger subtype, C1 (65% of positions), is generally intolerant to substitutions with aromatics being the most tolerated, whereas C2 (26.3%) tolerates any hydrophobic residue (Fig 4A). C1 positions tend to be involved in cysteine-specific functions, in particular disulphide bonds (Fig 4B) and various ligand binding interactions (Fig 4C). Conversely, the majority of C2 positions are buried (Fig 4D) and primarily utilise hydrophobic properties. Both subtypes also appear to be involved in interactions with aromatic residues via their π-orbitals (Orabi & English, 2016) (Fig 4E).

Another interesting set of subtype groups is those of the charged amino acids: negatively charged aspartate and glutamate and positive arginine and lysine. These visually split into 12 subtypes that fall into five broad categories (Fig 5A), with the frequency of charged amino acid positions falling in each category noted: selective for each of the two charge polarities (26.7% negative and 21.1% positive); selective for general polarity (7.83%); selection against negative charge (6.18%); and selection against proline only (12.2%), leaving 26.1% in rarer subtypes, permissive positions and outliers. These groups are quantitatively differentiated by the average electrostatic force they contribute to the protein (Fig 5B). The two groups that require charge tend to have the strongest forces, then the polar subtype group and the two selecting against properties have the weakest. A number of examples illustrate the typical roles of these different subtypes. Positions in the polar subtype group frequently interface with solvent (Fig 5C). Positive and negative positions together form ionic bonds (Fig 5D) and independently bind substrates such as DNA (Fig 5E) or ions (Fig 5F). Positions selecting against negative charge are often near negative charges that would repel them, either other residues in the protein or bound substrates such as RNA (Fig 5G) and, as is generally typical, positions selecting against proline occur where fold topology is important, particularly at the ends of secondary structures (Fig 5H) or in loops connecting them. The division of roles between subtypes is

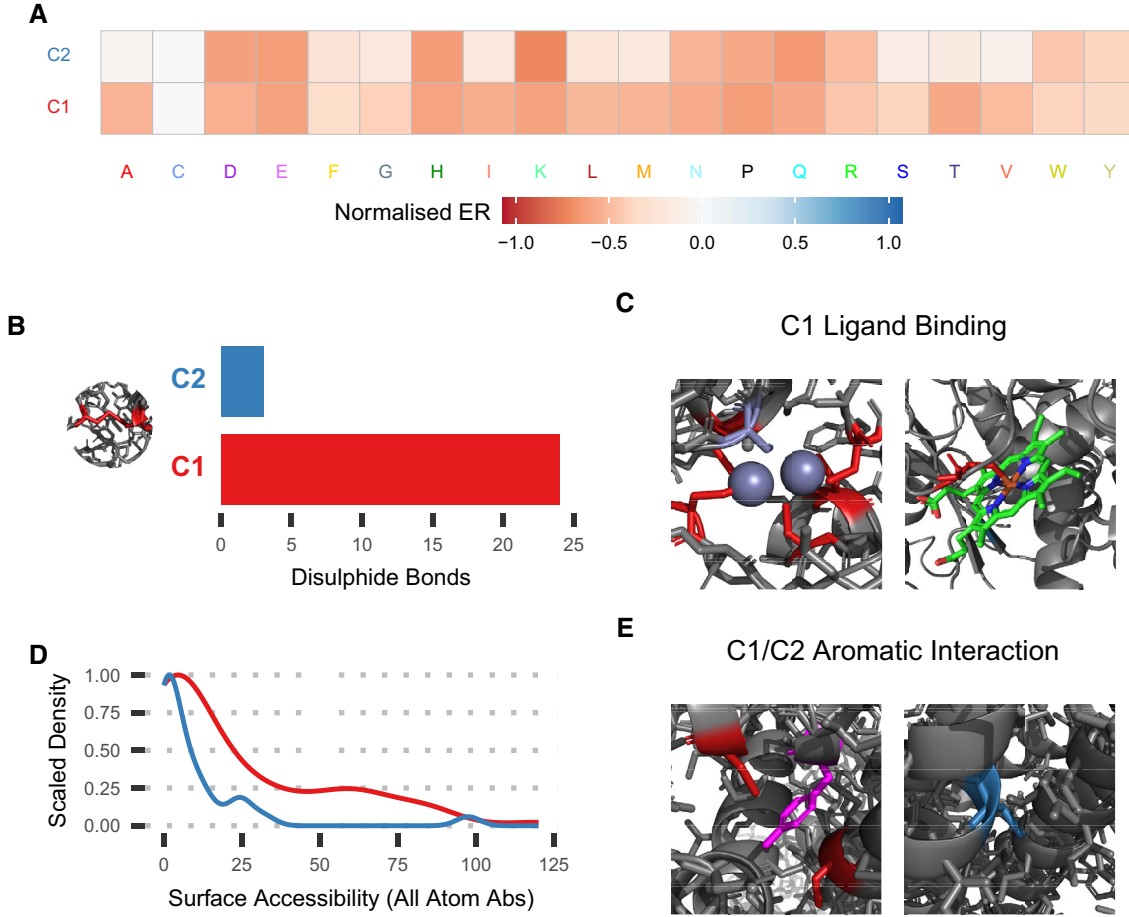

**Figure 4. Cysteine subtype examples.**

A   Mutational profiles of the two cysteine subtypes, C1 (red in other panels) and C2 (blue).
B   Number of positions of each subtype in a disulphide bond, based on FoldX prediction.
C   Examples of C1 positions involved in zinc ion (left, GAL4, PDB ID: 3COQ) and haem (right, CBS, PDB ID: 4L0D) ligand binding.
D   All atom absolute surface accessibility of cysteine subtypes.
E   Examples of C1 (left, NP, homology model on PDB ID: 2Q06) and C2 (right, CCR5, PDB ID: 6MET) interacting with aromatic groups.

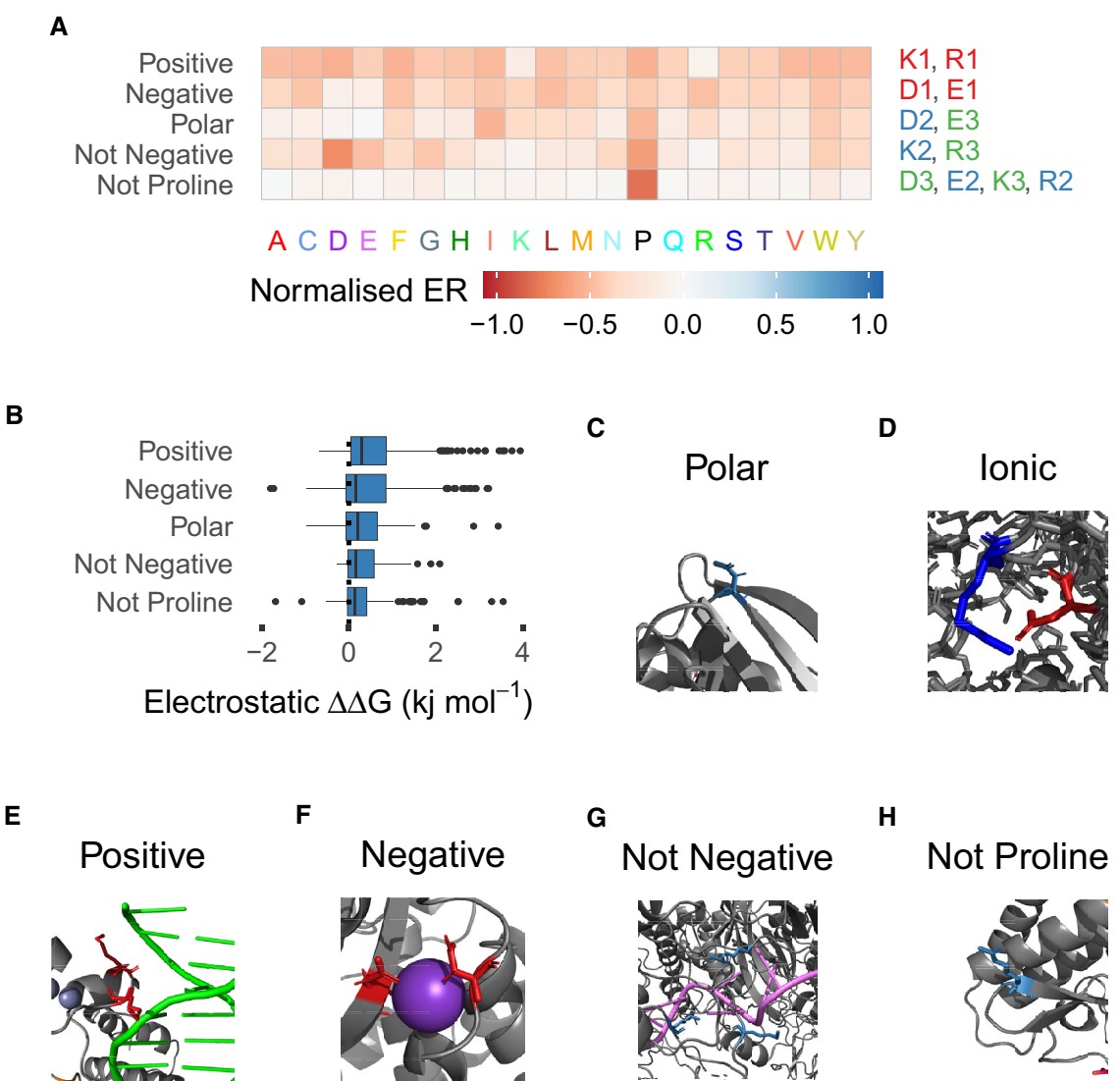

**Figure 5. Polar subtype examples.**

A  Mutational profiles of polar subtype groups. The group name is shown to the left of the profile and the subtypes that make it up to the right, coloured by subtype number and therefore frequency within positions of that amino acid.

B  Boxplots showing the distribution of the mean electrostatic component of FoldX substitution $\Delta\Delta G$ at each position of the subtype group. The box extends from $1^{st}$ to $3^{rd}$ quartile with a band at the median. The whiskers extend to the furthest point up to 1.5 times the interquartile range away from the nearest quartile, with any further points marked as outliers.

C  Polar position on the protein surface (TEM1, PDB ID: 1M40).

D  Example ionic interaction between positive and negative subtype positions (CBS, PDB ID: 4L0D).

E  Positive subtype position interacting with the DNA backbone (GAL4, PDB ID: 3COQ).

F  Negative subtypes binding an ion ligand (TEM1, PDB ID: 3COQ).

G  "Not negative" subtype positions surrounding bound RNA (PAB1, PDB ID: 6R5K).

H  "Not proline" position at the end of an alpha helix (APH3II, PDB ID: 1ND4).

not absolute but the majority of cases of each role examined in structural models fall into the appropriate subtype, which is also the case with other examples mentioned (see Dataset EV4).

The final example covers groups of subtypes requiring different sized hydrophobic amino acids, with one group of subtypes selecting for aromatics (F1, W1, Y1), one larger aliphatic amino acids (I2, L2, M2) and one small amino acids (A1, G2, P3; Fig 6A). The

frequency that positions of each amino acid fall into these groups varies a lot. For example, 36.5% of phenylalanine positions and 34.8% of tyrosine positions are in the "aromatic" group compared with 67.8% of tryptophan positions, suggesting tryptophan positions more commonly primarily utilise aromatic characteristics. The typical roles these subtypes have are illustrated by the average structural consequences of mutations at their positions, based on

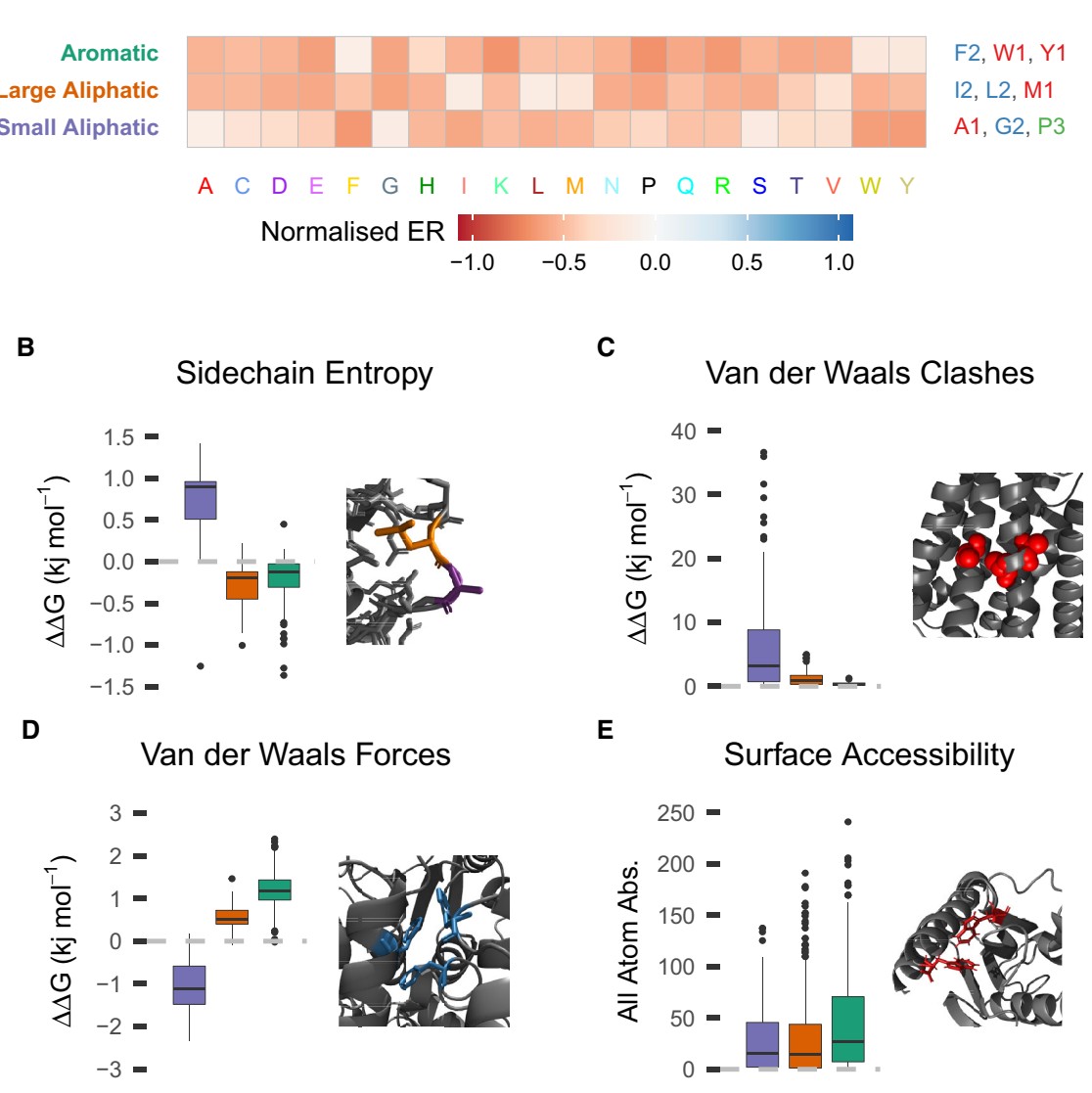

**Figure 6. Different sized aliphatic subtypes.**

A   Mutational profiles of size conscious aliphatic subtype groups. The groups are named on the left, coloured to match the boxplots in (B–E), and the constituent subtypes of each group are on the right, again coloured by subtype number.

B–D   Boxplots showing the distribution of mean FoldX substitution ΔΔ*G* across positions of each subtype group, for various FoldX terms. The box extends from 1st to 3rd quartile with a band at the median. The whiskers extend to the furthest point up to 1.5 times the interquartile range away from the nearest quartile, with any further points marked as outliers. (B) Sidechain entropy and an example showing why mutations at small aliphatic group positions increase entropy more. (C) Van der Waals clashes and an example showing alanine positions in a helix bundle demonstrating why clashes occur on mutation of small residues. (D) Van der Waals forces and an example showing internal aromatic groups making Van der Waals interactions.

E   Boxplot showing the distribution of all atom absolute surface accessibility between groups and an example of surface aromatic groups. The boxes and whiskers are defined in the same way as (B–D).

components of FoldX's substitution ΔΔ*G* predictions. Firstly, subtypes requiring small amino acids make a bigger contribution to entropy minimisation during folding, having fewer possible configurations and helping the protein fold efficiently (Fig 6B). They also frequently occur in cramped spaces within the proteins, resulting in clashes when mutated (Fig 6C). On the other hand, larger amino acids, both aromatic and aliphatic, contribute more Van der Waals forces (Fig 6D), highlighting the trade-off between Van der Waals forces, packing the correct folds and entropy minimisation when

selecting for amino acids in the hydrophobic core. Finally, aromatic positions are more likely to be surface accessible than either aliphatic position (Fig 6E, mean 16.8 Å² greater accessible surface area, one-tailed *t*-test: $P = 8.909 \times 10^{-6}$), including when the slightly polar tyrosine is excluded. This potentially occurs because the aromatic ring interacts somewhat with water through its delocalised π-orbitals (Slipchenko & Gordon, 2009). Together, these examples illustrate some properties and typical roles of subtypes and show that the subtypes have biological and structural features in specific

cases as well as in bulk statistics. In total, we have found 100 subtypes, characterised in Appendix Fig S8–S27 and described qualitatively with examples in Dataset EV4.

## Discussion

We have demonstrated the utility of combining deep mutational scanning studies for performing proteome scale analyses, even when the experiments differ between studies. While the results of these studies are strongly related to evolutionary conservation, they also provide quantitative information about rare variants that would require prohibitively large sample sizes to measure through sequencing of different species. As we demonstrate here, this makes them useful for analysing mutational profiles because these frequently include rare variants. While this dataset only covers a small slice of the protein universe, and so only forms the start of a reference landscape, many findings relate to common protein features and so we expect that it may apply to a wide range of other proteins. For example, fundamental features such as secondary structure folds, ionic bonding and ligand binding chemistry are similar across the proteome. On the other hand, features specific to the studied proteins will not necessarily generalise and many protein properties will not be represented, particularly those specific to underrepresented categories, for example prokaryotes or metabolic enzymes. In addition to changes in the mutational landscape driven by the addition of a growing number of positions, it is also important to note that the selection pressures imposed during DMS experiments can also vary. In the future, it will also be relevant to measure the extent by which such a joint analysis of DMS results may be influenced by changes in selection criteria.

The mutational landscape we describe clearly demonstrates the link between mutational profiles and biophysical properties across a range of different proteins. It is a potentially powerful tool for data-driven characterisation of a position's likely properties using mutational profiles or conversely estimation of mutational profiles from properties. This can be thought of as a continuous form of subtypes, where each position is described by a vector of properties, such as the principal components or UMAP transformation of its mutational profiles, and these vectors indicate biophysical properties of the position. These representations could also be useful for predictive models, especially if the landscape was developed with more data and more sophisticated methods linking it to properties.

The discrete classification of positions into amino acid subtypes provides similar power in a clearer form, providing a mapping between types of position and mutational profiles, as well as providing unique benefits such as the ability to more clearly quantify the range of diverse functions each amino acid fulfils and their frequencies. In this work, we identify 100 subtypes, including permissive positions of each amino acid but excluding outlier positions. The average ER scores for the subtypes were found to be typically either neutral (tolerating some mutations) or negative (not tolerating) and only rarely positive (preferring; see Fig 3C). This relates to the fact that mutations in these datasets generally destabilise rather than stabilise function, as seen in the example distributions in Fig 1C. We think this is in line with the biological expectation that mutations away from the natural protein sequence are more likely to destabilise the function of the protein than improve it.

One key aspect that depends on the joint analysis of multiple proteins is determining the differences in the frequency of these roles. As examples, we have discovered how often each amino acid is used in proteins for their most specific purposes (Fig 3E), including that cysteine positions seem to utilise cysteine-specific properties in 65% of positions or that charged residues only rely heavily on their charge in 21–27% of positions. The identification of amino acid subtypes, their characterisation and quantification of how often they are used in proteins is only possible when combining data from many proteins. This allows us to quantify patterns across proteins and highlight recurring patterns that are not apparent when considering each protein individually. For example, if we had studied 1,000 protein positions, which is larger than a typical single protein, we would have only been able to identify 17 subtypes (Appendix Fig S7). To allow for the future extension of this work, we make the DeepScanScape R package available, allowing others to apply our landscape and subtypes analysis to new DMS studies.

Subtype characterisation also breaks down typical roles of amino acids and their frequencies, which helps quantify the spread of functions across the proteome. The subtypes we identify include both amino acid-specific functions like ion binding and general roles, such as selection against proline or for hydrophobicity. This highlights another way to view some subtypes; they are shared between amino acids, and so an alternative approach would be to combine these subtypes, either after subtypes have been generated or by clustering all amino acids together. Such an analysis could help further quantify amino acid diversity by explicitly identifying positions where several amino acids share the same properties.

The relationship between deep mutational scanning and evolutionary conservation suggests that a similar mutational landscape analysis could be performed using sequence alignment data. We experimented with applying a conservation-based analysis to our dataset, based on positions SIFT4G score profile. The SIFT4G-based mutational landscape captures many of the same properties as that based on deep mutational scanning, although not the positioning of transmembrane protein domains (Appendix Fig S28). However, subtypes produced using the same algorithm on $\log_{10}$SIFT4G score profiles fail to capture many of the roles that ER base subtypes do. For example, disulphide bonds are shared evenly between the two SIFT4G-based cysteine subtypes, "not proline" type subtypes are not identified and aspartate positions binding ligands are not separated as clearly. In addition, ER-based subtypes average profiles are more differentiated from other subtypes of the same amino acid than SIFT4G-based subtypes (Appendix Fig S29, mean 0.13 greater cosine distance, one-tailed Mann-Whitney $U$-Test: $P < 2.2 \times 10^{-16}$). This suggests that conservation-based subtypes are a potentially powerful future direction, allowing coverage of a much larger portion of the proteome, but lack the resolution and detail provided by deep mutational scanning approaches. A method combining both data types could also be powerful.

Exploring how to combine and use this type of data will be important in future because many more deep mutational scanning experiments are expected. The most recent DepMap release (Meyers *et al*, 2017; Tsherniak *et al*, 2017; preprint: Dempster *et al*, 2019) suggests 7,293 genes are essential in at least one cell line and thus directly targetable by deep mutational scanning. This suggests data from 1,000s of genes could be available in future, ideally from more standardised experiments. While we have data from a diverse range

of proteins and have amassed as large a dataset as possible, we are ultimately limited to data from 6,291 positions, a very small slice of the proteome. Although the fact that our analysis largely relates to protein biochemistry rather than specific protein functions suggests our results may still be broadly applicable across the proteome, this is part of the reason we do not, and did not expect to, identify clear novel amino acid functions from our data; we only had access to data from a limited range of well-studied proteins. Potentially, a larger range of proteins would contain enough examples of rare, unstudied interactions to identify novel roles.

A larger dataset has potential for both extending this work and exploring other areas, for example a proteome-wide analysis of different types of substitution in different contexts (i.e. focussing on the mutant amino acid rather than the wild type). For our work, a larger dataset has the potential to expand the range of roles discovered, with rarer subtypes either completely missing or too rare to identify in our data. The 100-amino acid subtypes need to be considered a lower bound estimate of amino acid roles in proteins that can be derived in this way. A form of subtype we would expect to find with more data is post-translational modification sites such as phosphosites. However, we currently only cover 52 phosphosites (The UniProt Consortium, 2019), and these are not all necessarily conserved, active sites. These sites would potentially tolerate other phosphorylatable residues and phosphomimetic amino acids. Another potential outcome is dividing subtypes we have identified into more specific forms, for example splitting C1 into disulphide bonding and ligand binding positions. Thus, our estimates form lower bounds for subtype numbers, although with diminishing returns as you add more data. Finally, all the subtypes we identify are spread between at least 7 fairly dissimilar genes, suggesting we are not identifying roles specific to classes of protein, for example in active sites, which could be discovered with more data giving more examples of such specific functions.

Overall, our analysis shows three key points. Firstly, deep mutational scanning data can be combined from disparate studies into a meaningful dataset that relates to real biology. Secondly, the mutational landscape is a powerful tool for analysing protein positions, with rich relationships to biophysical properties, enabled by combining data from many proteins. And finally, positions of each amino acid can be broken down into typical subtypes using these profiles, allowing us to quantitatively map the diverse range of functions each amino acid plays across the proteome and the frequency of their usage.

# Materials and Methods

## Combining the data

First, we selected as many deep mutational scanning studies (Dataset EV1) as we could find that fulfilled the following criteria:

- Available data.
- Selection criteria matching the proteins' natural function.
- Scores that cannot be transformed to the standard $\log_2 \text{ER}_{\text{mut}} = \log_2 \frac{f^{\text{mut}}_{\text{post}}/f^{\text{mut}}_{\text{pre}}}{f^{wt}_{post}/f^{wt}_{pre}}$ form, where $f^{\text{mut/wt}}_{\text{pre/post}}$ is the frequency of variant mut or the wt before or after selection.

This included searching MaveDB (Esposito *et al*, 2019) and searching the literature. Each study was then processed into a standard state, with a complete set of variant scores for single substitutions at each position.

Some studies generated multiple mutations to individual sequences, without generating all single substitutions as well. In these cases, we used the average ER score of sequences including each substitution that was not directly tested, limiting the average to sequences with a maximum number of variants, depending on the deviation from known single substitution scores and resultant variant coverage (Appendix Fig S1, Dataset EV1). In studies with multiple comparable replicates, either direct replicates or in equivalent conditions, we averaged across conditions, and when multiple incomparable conditions were available, we chose the most representative of proteins' natural functions (Dataset EV1). Studies were then filtered if they had very poor correlation with SIFT4G scores, indicating unrealistic selection criteria or experimental oddities, or low coverage of substitutions at each position.

The scores for each study were transformed onto the standard scale, with a different transform required in each case (Dataset EV1), and normalised by dividing all scores by the median score of the lowest 10%. This threshold was chosen heuristically to encompass the typical ER scores of nonsense mutations. These scores were put together into a combined dataset, from which positions with < 15 of the 20 missense and nonsense substitutions were filtered and remaining missing data were imputed using the median of that substitution type (A → C, A → D,...) across all normalised scores. Some studies include variant codons and so measure synonymous variants' ER but where not measured synonymous substitutions were imputed to be 0 because 58% of study values were exactly 0 and 92% have absolute values less than 0.05.

We identified the best structural model for each protein in SWISS-MODEL (Waterhouse *et al*, 2018), selecting higher resolution and coverage where possible, as well as favouring experimental models over homology models. The models used are detailed in Dataset EV1.

## Analysing the mutational landscape

The combined mutational landscape data were annotated with additional biophysical data from a number of tools:

- SIFT4G – substitution effect scores (with a custom patch to output SIFT4G scores to 4.d.p).
- FoldX – substitution $\Delta\Delta G$ predictions.
- naccess – surface accessibility measurement.
- Porter5 – secondary structure predictions.

The different physical components of the FoldX $\Delta\Delta G$ predictions were averaged across all substitutions at a position in most analyses, giving a measure of the importance of different structural effects at that position. The UMAP (preprint: McInnes *et al*, 2018; Melville *et al*, 2020) and principal component dimensionality reductions were calculated and cross referenced with those factors.

## Identifying amino acid subtypes

We tried a number of methods to cluster amino acid positions, using various algorithms, distance metrics and profile formulations. Our

final algorithm, which performed by far the best, was as follows, applied independently to positions of each amino acid:

- Separate permissive positions ($|ER| < 0.4$ for all substitutions).
- Apply hierarchical clustering to the remaining positions, with average linkage and cosine distance to profiles consisting of PC2–PC20 (calculated across the whole mutational landscape).
- Use hybrid dynamic tree cutting (Langfelder *et al*, 2008) to assign positions to subtypes, using deepSplit = 0 or 1 depending on the amino acid.

We decided to use principal component space, with PC1 excluded, and cosine distance to remove the influence of mean ER score, which otherwise dominated clustering. Splitting positions by overall functional importance would also be an interesting analysis, but masks differentiation based on biophysical role, which is what we were interested in. Using PC2 to PC20 achieves this because PC1 captures the variation in mean ER. Using the cosine distance also helps because it measures the angle between the vectors formed by two profiles in multidimensional space, which is independent of their overall magnitude. However, using this metric means the distance between low magnitude positions is very noisy, since their "direction" is largely random. To combat this, we manually separate these positions before clustering.

The deepSplit parameter to the dynamic tree cutting algorithm determines how much to split the dendrogram. We vary this parameter between amino acids, increasing it to 1 for amino acids that appear to under-split using 0, based on profile consistency in the resultant subtypes. This is potentially partially required due to the amount of data we have, and more data could allow us to optimise this parameter across the dataset as a whole.

Subtypes were characterised using the averages of the various statistics and metrics we previously annotated the mutational landscape with, as well as their average ER score profiles.

The number of subtypes identified with variable numbers of positions was determined by applying the clustering algorithm to increasingly large subsets of 100 positional shuffles of the main dataset. 1,000 positions were initially included in each and then increased in increments of 200.

## Data availability

The datasets and computer code produced in this study are available in the following databases:

- Combined landscape data: Dataset EV2.
- Subtype assignments: Dataset EV3.
- Dataset processing, landscape analysis and clustering code: GitHub (https://github.com/allydunham/aa_subtypes).
- DeepScanScape R Package: GitHub (https://github.com/allydunham/deepscanscape).

DeepScanScape allows users to process new DMS studies using the methods presented here and analyse them based on the deep landscape dataset. This lets you identify unusual properties of studies and positions, visualise data on the deep landscape and assign amino acid subtypes to positions to predict potential properties.

**Expanded View** for this article is available online.

## Acknowledgements

## Author contributions
Project design: ASD and PB. Dataset curation: ASD. Data analysis: ASD. Manuscript preparation: ASD and PB. Project supervision and management: PB. Funding acquisition: PB.

## Conflict of interest
The authors declare that they have no conflict of interest.

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
