## [Review Process File · Molecular Systems Biology]

Exploring amino acid functions in a deep mutational landscape

Alistair Dunham and Pedro Beltrao

DOI: [10.15252/msb.202110305](https://doi.org/10.15252/msb.202110305)

Corresponding author: Pedro Beltrao (pbeltrao@ebi.ac.uk)

Review Timeline:

Transfer from Review Commons:	19th Feb 21
Editorial Decision:	23rd Feb 21
Revision Received:	18th May 21
Editorial Decision:	23rd Jun 21
Revision Received:	29th Jun 21
Accepted:	30th Jun 21

Editor: Maria Polychronidou

Transaction Report:

Review
COMMONS

(Note: This manuscript was transferred to Molecular Systems Biology following peer review at Review Commons. With the exception of the correction of typographical or spelling errors that could be a source of ambiguity, letters and reports are not edited. Depending on transfer agreements, referee reports obtained elsewhere may or may not be included in this compilation. Referee reports are anonymous unless the Referee chooses to sign their reports.)

Review #1

1. How much time do you estimate the authors will need to complete the suggested revisions:

Estimated time to Complete Revisions (Required)

(Decision Recommendation)

Between 1 and 3 months

2. Evidence, reproducibility and clarity:

Evidence, reproducibility and clarity (Required)

This manuscript provides an interesting approach to analyzing existing large datasets of mutational protein landscapes. Briefly, an amino acid can play different roles within proteins depending on its chemical environment/context, and the roles played at a position affect a position's mutational profile. Recent technological advances have facilitated high-throughput experimental characterizations of mutational profiles for a number of proteins.

Here, the authors seek to identify, classify, and analyze the roles individual amino acids can exhibit using data from 28 of these recent experimental characterizations (with mutational landscapes of 30 genes using 33 deep mutational scans). The strength of the work is the approach, which involved transforming and normalizing results from multiple fitness landscape studies so their results can be combined and directly compared. The authors then visualized mutational landscapes through dimensionality reduction techniques (UMAP, PCA) and analyzed relationships between and derived patterns for biophysical properties of protein positions and their placement in UMAP space. Finally, they defined subtypes for each amino acid to map role diversity and analyzed several subtypes to greater depth to better understand their specific properties

While I am intrigued by the goals and findings of the manuscript, and appreciate the holistic approach to understanding better the biophysical properties of amino acids, I did struggle with many aspects of the work in its current form:

1. It is not clear that any of the relationships or subtypes can be used to predict or otherwise understand proteins that have not already had mutational fitness landscapes analyzed experimentally. While perhaps this is for future work, the impact of the current work is reduced without such an example.

2. Many of the key findings - subtypes and examples of what they might mean for specific residues of the model proteins - are difficult to parse because so much is not explained in the main text. This is true throughout the manuscript.

a. For example, SIFT scores are introduced, and a definition of SIFT scores is given, but

the interpretation of SIFT scores is not clear for readers unfamiliar with SIFT scores. A brief description of interpretation SIFT scores (such as theoretical range, what the edges of the range represent) could be beneficial.

b. Another example: it is stated that "all studies span UMAP space so the dimensionality reduction demonstrates a lack of batch effects" but it is unclear how a position would not span UMAP space if that position was used in the creation of UMAP space.

c. A third, less complex example: In figure 3B, what are P and O? Permissive and Outlier? If so, state this. In Figure 3C, state that the larger graph is a Pearson correlation and the smaller heatmaps are the ER. In the text, put this in context - what does this tell the reader?

3. On a related note, to point 2, this is at its core a data science paper, heavy on statistics. However, the goal (presumably) is for the researchers who experimentally mutate proteins to learn from it. As written, the heavy use of statistical terms and other field-specific jargon without additional explanation or context makes it difficult to parse by any who are not experts in those areas. There were too many examples to list but a few are here:

a. Porter5 is first mentioned on page 9 without context, and then again on page 17 with a little more context, and then again in Figure S8 caption. Before its first use, there should be a brief sentence describing what Porter5 is and why it was chosen compared to any other options for secondary structure.

b. It is stated that: "strong role of mean normalised ER and its link to overall evolutionary conservation at a position (figure 2B)." There is nothing in this figure that highlights this fact to me, though I see the UMAP2 vs UMAP1 data. This needs to be more clearly labeled, and/or better explained in the text. (This goes back to the SIFT scores and their use.)

c. Despite all of the technical terms, the subject characterizations of Figure 3E fall flat due to the inclusion of so many statistical calculations. What are the numbers that correlate to "most selective" or "permissive"? Include or reference the supplemental figures that show this data, and give more explanation and/or context in the main text.

d. The details on the subtypes are included in the supplemental figures, but a more detailed explanation would be helpful. For example, it would even help to supply a more succinct explanation that the 1 subtype is not the same for each amino acid and that each number subtype can have different physical meaning for an amino acid.

4. In the discussion, the authors refer to their work as a way to check amino acid functions across the proteome, but it's a small sample of proteins selected without relation to each other, and it is not clear that these cover the proteome's multidimensional space adequately, nor is any evidence given to this effect. If this is too broad of a claim (as I suspect), it is still useful to have it dialed back to the fact that this is the start of such a reference database. To that end, some explanation is given as to why these proteins are chosen, but how does that connect to understanding the proteome? A large number of this study's proteins are derived from *H. sapiens*, with the

remainder dominated by bacteria and fungi; what of other organisms?

5. On a related note to point 4, whenever exploring fitness landscapes, the "fitness" definition is extremely important. For example, enzyme activity varies significantly across conditions, and an amino acid that is permissive under the conditions tested may not be so under an alternative (still useful) set of conditions. This was nicely acknowledged on page 6 when introducing the approach, using membrane proteins as the example, but I still struggle to see how the authors can be confident in the results without a much larger data set. Unknown nuances of this variety mean that the findings in this work are only good for explanations of proteins of the types profiled, and in the biophysical environments that were used to generate the original landscapes. This limits the utility of the findings. Note that I am not saying the results are not useful at all, but the authors may wish to make clear HOW to use the findings appropriately.

****Minor Comments:****

1. The common score is denoted the enrichment ratio (ER) score and represents the enrichment of a variant during selection relative to the change observed in wild type, such that positive scores are advantageous and negative ER scores are deleterious, but in nearly all the figures it appears that ER is always less than 0, except perhaps on the "Not Proline" heatmap in Figure 3. Presumably there are very few advantageous mutations compared to WT in the datasets used, but if this is the reason, it should be explicitly discussed.

2. "either selected the most appropriate condition" - need some reference to what was done to decide what was appropriate

3. Section "A unified deep mutational landscape": "The scores from these results were correlated" - r^2 less than 0.5 seem weak for correlation.

4. Figure 1B: Need x-axis for Raw graph

5. Figure 2C: the 6 is cut off on the x-axis of the graph

6. Figure 2 caption is light gray instead of black

7. Throughout Fig 4 - 6, it was not always clear that the y-axis was a specific subtype for that amino acid. That needs to be clearer

8. Section "Mapping the diversity of amino acid subtypes": "Permissive positions have low ER scores for all substitutions" - Why do permissive positions have low ER? It seems that all the mutations highlighted in the paper are worse than wild-type.

9. How does this approach handle post-translational modifications and their impact?

3. Significance:

Significance (Required)

The advance here is the combining and normalizing of mutational datasets. As noted above, however, it is not obvious to me that in so doing, new insights are pulled from the data. Toward Major comment 4, can the authors perhaps point to findings that give an insight that required the combined power of the 30 datasets and does not come out of analysis of a single protein's mutational landscape?

Nonetheless, the work is unique among the fitness landscape field, and it could be transformative to combine datasets in this way.

The audience of the work as written is probably data scientists, and this needs to change so that the findings can be employed by those who study protein evolution or engineering (the areas that I work in, as a wet-lab experimentalist). Indeed, if written for this broader audience - or, even better, by suggesting ways to design mutational landscape studies such that the results could be included in future iterations of this analysis - I think that the work could be of high impact to our understanding of protein structure and function from a biophysical point of view.

Review #2

1. How much time do you estimate the authors will need to complete the suggested revisions:

Estimated time to Complete Revisions (Required)

(Decision Recommendation)

Between 1 and 3 months

2. Evidence, reproducibility and clarity:

Evidence, reproducibility and clarity (Required)

This was a very interesting study, integrating a large number of deep mutational scanning datasets to explore the functional properties of different amino acids. It is important to emphasize that it is very much an exploratory study - there is unfortunately not a lot in terms of novel biological insight.

I have no major concerns in regards to methodology, but do note a number of points:

- The UMAP projections presented throughout this paper are not as informative as they could be. This is mostly due to high density of overlapping opaque points, grouped in such a way that 'only last plotted is seen'. This could be improved by either using smaller points or increasing transparency. An alternative is to divide the plot into small grid squares, then colour based on the average point value in each square (on a two-

colour scale). Shade by density of points.

- What is the threshold for similar conditions in a DMS experiment to be comparable, and thus averaged, rather than be regarded as different conditions?

- The assumption that the bottom 10% of DMS results represent null mutations is not necessarily sound, not all DMS studies look at nonsense mutations, so the results may be slightly skewed in those that do vs those that do not. I would like to see some validation of this approach.

- Figure 2A. Hemagglutinin also has a transmembrane helix, does the observed pattern hold up here as well?

- Page 10, possible typo: "This may be because they tend to occur in positions that at least require hydrophobicity, such as the core of the protein, and therefore at least strongly hydrophobic substitutions are selected against"

I think they mean that non-hydrophobic substitutions are selected against in the core of the protein?

- Possible additional analysis: Estimate of the total number of subtypes by starting with a small number of proteins in the analysis and seeing how many more are identified with each protein added back in. The function should converge towards the estimated total number.

- The authors state throughout that they are using SIFT, but we see in the methods that it is actually SIFT4G. They can be quite different in their results, so I think it would be better to call them SIFT4G scores throughout.

- What does AP stand for in Fig 3?

- "We filtered positions with scores for less than 15 of the possible 20 nonsynonymous substitutions (including nonsense) to focus on positions with sufficient data and imputed remaining missing data" How much data are you losing with this strict criteria? As many DMS studies only focus on amino acid substitutions possible from single nucleotide changes, you will lose all of these, so I am curious whether you could still get meaningful amino acid subtypes when considering such datasets.

- "31 of 66 positions covered by multiple studies were assigned to the same subtype". First, this seems like a low number of positions, given that it presumably includes both HSP90 and UBI datasets, unless these have very low coverage. Second, it seems like fairly mediocre consistency - it is statistically significant, but suggests that an accurate subtype cannot be determined for more than half of positions.

- BLOSUM62 is an amino acid substitution matrix - I don't understand how it can provide scores for synonymous changes, so I must not be understanding how it is used here (unless you mean score of 0 for a synonymous change).

- Do synonymous changes always have ER=0, because differences in fitness due to synonymous differences are sometimes observed in DMS studies?

3. Significance:

Significance (Required)

I do believe that the paper is both interesting and useful, and the authors do stress several times that their results shouldn't be taken as conclusive, since addition of more data will likely increase the number of amino acid categories they identify. Given this, it seems to be presenting the clustering methodology and giving examples of uses rather than analysis of meaningful results. (This is actually in line with their stated objectives, I just thought they could have gone further.)

Review Commons Comments

General Changes

Our reading of the comments of the reviewers is that they both appreciate the work as an original effort to merge together and jointly analyze the space of deep mutational protein experiments. Reviewer 1 wrote that this work is “unique among the fitness landscape field, and it could be transformative to combine datasets in this way”, while reviewer 2 believes “that the paper is both interesting and useful”. Both reviewers expressed some concern that the manuscript is very jargon heavy and oriented towards a more data science technical audience. There is also a general concern that those scientists working on protein variation might not be able to make the most use of this work.

If MSB would be interested in receiving a revised version of the manuscript we would focus our revision in addressing these concerns. One of the main efforts we would make would be to make available a new software tool that could be used to annotate and analyse any deep mutational scanning map. This would add value to anyone wanting to make use of the research presented in this current manuscript for application of their protein of interest and any future DMS study. A second aspect of emphasis would be generally on clarity of the manuscript both to reduce the technical jargon and to highlight what are the key biological insights that are presented and why they could only be observed by the combined analysis of several DMS maps. Specifically we would aim to:

- Create a tool that scientists working on new mutational studies could use to analyse new and augment new experimental data with respect to:
 - Automatically analysing a new DMS dataset to assign subtypes to new positions and offer potential characterisation/explanations to what these represent for their protein under study
 - Compare statistical distributions of their protein/study to previous studies, which would indirectly offer a sense of the quality of a newly generated map.
- Emphasise the new biological insights and significance of the work that was achieved with the combined dataset, especially showing the findings that could not be found with single studies alone
- Clarify and simplify the technical aspects of the work.

Reviewer 1

Evidence, reproducibility and clarity

This manuscript provides an interesting approach to analyzing existing large datasets of mutational protein landscapes. Briefly, an amino acid can play different roles within proteins depending on its chemical environment/context, and the roles played at a position affect a position's mutational profile. Recent technological advances have facilitated high-throughput experimental characterizations of mutational profiles for a number of proteins.

Here, the authors seek to identify, classify, and analyze the roles individual amino acids can exhibit using data from 28 of these recent experimental characterizations (with mutational landscapes of 30 genes using 33 deep mutational scans). The strength of the work is the approach, which involved transforming and normalizing results from multiple fitness landscape studies so their results can be combined and directly compared. The authors then visualized mutational landscapes through dimensionality reduction techniques (UMAP, PCA) and analyzed relationships between and derived patterns for biophysical properties of protein positions and their placement in UMAP space. Finally, they defined subtypes for each amino acid to map role diversity and analyzed several subtypes to greater depth to better understand their specific properties

While I am intrigued by the goals and findings of the manuscript, and appreciate the holistic approach to understanding better the biophysical properties of amino acids, I did struggle with many aspects of the work in its current form:

Reading the points below from the reviewer, we believe quite a large part of these have to do with how we present the analyses and results. We will consider carefully how to present this work to a wider audience by introducing better concepts, analyses and results.

1. It is not clear that any of the relationships or subtypes can be used to predict or otherwise understand proteins that have not already had mutational fitness landscapes analyzed experimentally. While perhaps this is for future work, the impact of the current work is reduced without such an example.

We agree with the reviewer that this would have a broader impact if it was easily extended to future studies. We will therefore develop an analysis tool to go along with this manuscript that will allow other scientists to analyse a new dataset. This will allow them to integrate a new dataset with previous datasets, generating annotations and comparisons. We believe this would also serve as a form of quality control besides providing a rich set of annotations for their protein of interest. This will be a central focus of our revised manuscript that we think will raise the impact of the study.

2. Many of the key findings - subtypes and examples of what they might mean for specific residues of the model proteins - are difficult to parse because so much is not explained in the main text. This is true throughout the manuscript.

As for a few of the other comments of the reviewer, we will make a strong effort to clarify how the analyses were made and to explain the outcomes in ways that can be understood by a wider audience that is not familiar with the approaches.

a. For example, SIFT scores are introduced, and a definition of SIFT scores is given, but the interpretation of SIFT scores is not clear for readers unfamiliar with SIFT scores. A brief description of interpretation SIFT scores (such as theoretical range, what the edges of the range represent) could be beneficial.

As above, we will clarify for a wider audience.

b. Another example: it is stated that "all studies span UMAP space so the dimensionality reduction demonstrates a lack of batch effects" but it is unclear how a position would not span UMAP space if that position was used in the creation of UMAP space.

As above, we will clarify for a wider audience.

c. A third, less complex example: In figure 3B, what are P and O? Permissive and Outlier? If so, state this. In Figure 3C, state that the larger graph is a Pearson correlation and the smaller heatmaps are the ER. In the text, put this in context - what does this tell the reader?

As above, we will clarify for a wider audience.

3. On a related note, to point 2, this is at its core a data science paper, heavy on statistics. However, the goal (presumably) is for the researchers who experimentally mutate proteins to learn from it. As written, the heavy use of statistical terms and other field-specific jargon without additional explanation or context makes it difficult to parse by any who are not experts in those areas. There were too many examples to list but a few are here:

As for point 2a, 2b, and 2c, the following points relate to clarity and presentation of the results. We agree with the reviewer that the current version is very technically minded and needs to be made more accessible.

a. Porter5 is first mentioned on page 9 without context, and then again on page 17 with a little more context, and then again in Figure S8 caption. Before its first use, there should be a brief sentence describing what Porter5 is and why it was chosen compared to any other options for secondary structure.

As above, we will clarify for a wider audience.

b. It is stated that: "strong role of mean normalised ER and its link to overall evolutionary conservation at a position (figure 2B)." There is nothing in this figure that highlights this fact to me, though I see the UMAP2 vs UMAP1 data. This needs to be more clearly labeled, and/or better explained in the text. (This goes back to the SIFT scores and their use.)

As above, we will clarify for a wider audience.

c. Despite all of the technical terms, the subject characterizations of Figure 3E fall flat due to the inclusion of so many statistical calculations. What are the numbers that correlate to "most selective" or "permissive"? Include or reference the supplemental figures that show this data, and give more explanation and/or context in the main text.

As above, we will clarify for a wider audience.

d. The details on the subtypes are included in the supplemental figures, but a more detailed explanation would be helpful. For example, it would even help to supply a more succinct explanation that the 1 subtype is not the same for each amino acid and that each number subtype can have different physical meaning for an amino acid.

As above, we will clarify for a wider audience.

4. In the discussion, the authors refer to their work as a way to check amino acid functions across the proteome, but it's a small sample of proteins selected without relation to each other, and it is not clear that these cover the proteome's multidimensional space adequately, nor is any evidence given to this effect. If this is too broad of a claim (as I suspect), it is still useful to have it dialed back to the fact that this is the start of such a reference database. To that end, some explanation is given as to why these proteins are chosen, but how does that connect to understanding the proteome? A large number of this study's proteins are derived

from *H. sapiens*, with the remainder dominated by bacteria and fungi; what of other organisms?

We certainly agree with the reviewers concern that this is still a small sample of any proteome and we have tried already to discuss the fact that this is a starting point to be updated as new deep mutational studies are generated. We will further expand on this in a revised version of the discussion as per the suggestion.

5. On a related note to point 4, whenever exploring fitness landscapes, the "fitness" definition is extremely important. For example, enzyme activity varies significantly across conditions, and an amino acid that is permissive under the conditions tested may not be so under an alternative (still useful) set of conditions. This was nicely acknowledged on page 6 when introducing the approach, using membrane proteins as the example, but I still struggle to see how the authors can be confident in the results without a much larger data set. Unknown nuances of this variety mean that the findings in this work are only good for explanations of proteins of the types profiled, and in the biophysical environments that were used to generate the original landscapes. This limits the utility of the findings. Note that I am not saying the results are not useful at all, but the authors may wish to make clear HOW to use the findings appropriately.

While we agree with the reviewer that the selection criteria used for the experiments will impact on the results it remains unclear whether it will strongly impact on the clustering outcomes we present here. Most likely, the selection criteria will shift some positions of a given protein across different clusters but not strongly impact on the overall clustering itself. It will certainly not impact on the key idea presented of using the approaches developed here for further expansions of the deep mutational approach to other proteins and assays. We will investigate in the current dataset if different selection pressures tend to be strongly influencing where the protein positions tend to partition within the merged space of mutational outcomes. We believe this will provide some evidence in favour or against the hypothesis of the reviewer.

Minor Comments

We will take into account these useful minor comments in the revised version.

1. The common score is denoted the enrichment ratio (ER) score and represents the enrichment of a variant during selection relative to the change observed in wild type, such that positive scores are advantageous and negative ER scores are deleterious, but in nearly all the figures it appears that ER is always less than 0, except perhaps on the "Not Proline" heatmap in Figure 3. Presumably there are very few advantageous mutations compared to WT in the datasets used, but if this is the reason, it should be explicitly discussed.

2. "either selected the most appropriate condition" - need some reference to what was done to decide what was appropriate

3. Section "A unified deep mutational landscape": "The scores from these results were correlated" - r^2 less than 0.5 seem weak for correlation.

4. Figure 1B: Need x-axis for Raw graph

5. Figure 2C: the 6 is cut off on the x-axis of the graph

6. Figure 2 caption is light gray instead of black

7. Throughout Fig 4 - 6, it was not always clear that the y-axis was a specific subtype for that amino acid. That needs to be clearer

8. Section "Mapping the diversity of amino acid subtypes": "Permissive positions have low ER scores for all substitutions" - Why do permissive positions have low ER? It seems that all the mutations highlighted in the paper are worse than wild-type.

9. How does this approach handle post-translational modifications and their impact?

Significance

The advance here is the combining and normalizing of mutational datasets. As noted above, however, it is not obvious to me that in so doing, new insights are pulled from the data. Toward Major comment 4, can the authors perhaps point to findings that give an insight that required the combined power of the 30 datasets and does not come out of analysis of a single protein's mutational landscape?

We fully agree with the reviewer and we will make an effort to highlight the discoveries that could only have been made by combining the datasets in this way and not be possible to derive with any single dataset alone.

Nonetheless, the work is unique among the fitness landscape field, and it could be transformative to combine datasets in this way.

We appreciate the positive remark.

The audience of the work as written is probably data scientists, and this needs to change so that the findings can be employed by those who study protein evolution or engineering (the areas that I work in, as a wet-lab experimentalist). Indeed, if written for this broader audience - or, even better, by suggesting ways to design mutational landscape studies such that the results could be included in future iterations of this analysis - I think that the work could be of high impact to our understanding of protein structure and function from a biophysical point of view.

We will work to broaden the audience of this work by improving the clarity of the manuscript and primarily by developing a computational tool that scientists that are not versed in computational analysis could use to analyse any deep mutational dataset. This will allow them to annotate a new dataset and compare it with a compilation of other studies. We believe this will be a powerful addition to the field.

Reviewer 2

Evidence, reproducibility and clarity

This was a very interesting study, integrating a large number of deep mutational scanning datasets to explore the functional properties of different amino acids. It is important to emphasize that it is very much an exploratory study - there is unfortunately not a lot in terms of novel biological insight.

As per the response to reviewer 1, we think we can better highlight which findings are novel and only possible by the combined dataset. We also think that we will increase the impact and novelty of this study by providing a tool that other scientists can use in the analysis of any DMS dataset. We believe that revising the manuscript in this way will provide evidence to this reviewer that there is a strong advance being made here.

I have no major concerns in regards to methodology, but do note a number of points:

- The UMAP projections presented throughout this paper are not as informative as they could be. This is mostly due to high density of overlapping opaque points, grouped in such a way that 'only last plotted is seen'. This could be improved by either using smaller points or increasing transparency. An alternative is to divide the plot into small grid squares, then colour based on the average point value in each square (on a two-colour scale). Shade by density of points.

We will revise the presentation of the UMAP projections for clarity, We will try the suggestion of the reviewer and other approaches.

- What is the threshold for similar conditions in a DMS experiment to be comparable, and thus averaged, rather than be regarded as different conditions?

We will clarify in a revised version.

- The assumption that the bottom 10% of DMS results represent null mutations is not necessarily sound, not all DMS studies look at nonsense mutations, so the results may be slightly skewed in those that do vs those that do not. I would like to see some validation of this approach.

We will analyse this concern and study potential skews of the assumption used. We don't expect this will have a strong impact on the results.

- Figure 2A. Hemagglutinin also has a transmembrane helix, does the observed pattern hold up here as well?

We will perform the analysis as suggested.

- Page 10, possible typo: "This may be because they tend to occur in positions that at least require hydrophobicity, such as the core of the protein, and therefore at least strongly hydrophobic substitutions are selected against"

I think they mean that non-hydrophobic substitutions are selected against in the core of the protein?

We will revise the typo.

- Possible additional analysis: Estimate of the total number of subtypes by starting with a small number of proteins in the analysis and seeing how many more are identified with each protein added back in. The function should converge towards the estimated total number.

This is an interesting and useful suggestion, we will perform the analysis. This will also highlight the findings that could only be derived from a larger combination of datasets.

·The authors state throughout that they are using SIFT, but we see in the methods that it is actually SIFT4G. They can be quite different in their results, so I think it would be better to call them SIFT4G scores throughout.

We will revise this.

·What does AP stand for in Fig 3?

We will revise this for clarity.

·"We filtered positions with scores for less than 15 of the possible 20 nonsynonymous substitutions (including nonsense) to focus on positions with sufficient data and imputed remaining missing data" How much data are you losing with this strict criteria? As many DMS studies only focus on amino acid substitutions possible from single nucleotide changes, you will lose all of these, so I am curious whether you could still get meaningful amino acid subtypes when considering such datasets.

We don't think we lose much but we will provide explicit numbers.

·"31 of 66 positions covered by multiple studies were assigned to the same subtype". First, this seems like a low number of positions, given that it presumably includes both HSP90 and UBI datasets, unless these have very low coverage. Second, it seems like fairly mediocre consistency - it is statistically significant, but suggests that an accurate subtype cannot be determined for more than half of positions.

Both of the studies have low coverage so the total number of positions is modest because of this. While we agree that the consistency of subtypes is not high, some of the subtypes can be quite similar and we think a substantial amount of diverging classification will be among very similar subtypes. In a revised version we will better quantify these changes in classification in regards also to the similarity of the subtypes.

·BLOSUM62 is an amino acid substitution matrix - I don't understand how it can provide scores for synonymous changes, so I must not be understanding how it is used here (unless you mean score of 0 for a synonymous change).

We think this is a misunderstanding from the part of the reviewer and we will clarify this point in a revised version.

·Do synonymous changes always have $ER=0$, because differences in fitness due to synonymous differences are sometimes observed in DMS studies?

In the studies used here, the synonymous changes are almost always close to zero. We will expand on this point in the full response and can also better discuss this in the discussion section.

Significance

I do believe that the paper is both interesting and useful, and the authors do stress several times that their results shouldn't be taken as conclusive, since addition of more data will likely increase the number of amino acid categories they identify. Given this, it seems to be presenting the clustering methodology and giving examples of uses rather than analysis of

meaningful results. (This is actually in line with their stated objectives, I just thought they could have gone further.)

We appreciate that both reviewers understood the work and find it generally a useful addition to the field. As both reviewers have noted, this is a first attempt that we think of as an original piece of work that will be updated as new data is generated. However, we also agree that we can attempt to raise the impact of this. We think the addition of the software tool to analyse new datasets, generate annotations and comparisons with previous datasets will be an important improvement and be of much wider use.

RE: MSB-2021-10305, Exploring amino acid functions in a deep mutational landscape

Thank you for submitting your manuscript to Molecular Systems Biology. I have now read the manuscript and your preliminary point-by-point response to the comments of the Review Commons reviewers. I would like to invite you to submit your revised manuscript to Molecular Systems Biology.

Overall, we agree with the reviewers that the combined analyses of multiple deep mutational scanning datasets seem relevant. The concerns raised by the reviewers seem relatively straightforward to address and I think that your revision plan sounds promising. In particular, providing an analysis tool for future use is a good idea. Moreover, clarifying the advance and highlighting why this approach and the combined DMS datasets are key for deriving the presented conclusions and making the study more accessible to a broad audience will enhance the impact of the work.

The eventual acceptance of the study will depend on how well the issues raised by the referees have been addressed. As you might already know, our editorial policy allows in principle a single round of major revision, and it is therefore essential to provide responses to the reviewers' comments that are as complete as possible.

Please find below a point by point response to the reviewers' concerns, our responses in blue.

Reviewer 1

Evidence, reproducibility and clarity

This manuscript provides an interesting approach to analyzing existing large datasets of mutational protein landscapes. Briefly, an amino acid can play different roles within proteins depending on its chemical environment/context, and the roles played at a position affect a position's mutational profile. Recent technological advances have facilitated high-throughput experimental characterizations of mutational profiles for a number of proteins.

Here, the authors seek to identify, classify, and analyze the roles individual amino acids can exhibit using data from 28 of these recent experimental characterizations (with mutational landscapes of 30 genes using 33 deep mutational scans). The strength of the work is the approach, which involved transforming and normalizing results from multiple fitness landscape studies so their results can be combined and directly compared. The authors then visualized mutational landscapes through dimensionality reduction techniques (UMAP, PCA) and analyzed relationships between and derived patterns for biophysical properties of protein positions and their placement in UMAP space. Finally, they defined subtypes for each amino acid to map role diversity and analyzed several subtypes to greater depth to better understand their specific properties

While I am intrigued by the goals and findings of the manuscript, and appreciate the holistic approach to understanding better the biophysical properties of amino acids, I did struggle with many aspects of the work in its current form:

1. It is not clear that any of the relationships or subtypes can be used to predict or otherwise understand proteins that have not already had mutational fitness landscapes analyzed experimentally. While perhaps this is for future work, the impact of the current work is reduced without such an example.

We agree with the reviewer that it would be useful to extend the work to allow future studies to be incorporated and more broadly to extend the impact of the work presented here. To address this we have developed an analysis tool that we are making available with the manuscript (<https://github.com/allydunham/deepscanscape>). This tool allows others to integrate a new dataset with previous datasets, generating annotations and comparisons.

This new tool can be run in two settings:

- Comparison of a new study against the collection described in this manuscript: This allows anyone that has performed a new study to make predictions about how each position in their study matches one of the subtypes presented here. Based on this we can also make predictions about some of the biophysical properties of the annotated predictions which can add substantial value for cases when the structure is not yet available for the studied protein (see below).
- The second mode of analysis would be to analyse a new or extended compilation of DMS studies. This allows someone to normalise a new compilation of studies and to perform the same clustering and amino-acid subtype identification that we have performed here. This will facilitate the expansion of the deep mutational landscape to larger sets of proteins. All of the code is accessible such that more advanced users

can also adapt it and extend based on new ideas.

To exemplify the first mode of usage of this tool we have analysed a DMS of the SARS-CoV-2 Spike protein (Starr et al., 2020) which was not previously included in our analysis. The newly developed package can attempt to annotate each of the positions of a new dataset to one of the subtypes defined in our study. Subtypes are assigned based on the cosine distance to the mean profiles of each of the subtypes. Positions that are similar distances to multiple clusters are labeled ambiguous and those too far from all subtypes as outliers. To test the accuracy of such assignments we used this method on the positions that are already mapped in the landscape dataset. This led to an accuracy of 93% and micro and macro F1 scores of ≈ 0.75 . Based on this approach we used the newly developed tool to assign amino-acid subtypes to positions from the Spike protein.

Based on subtype assignments the user can gain insights based on the information we provide in this manuscript but we then can also provide the user with predictions for several biophysical properties based on the average values of each amino-acid subtype in the compilation. We compared these predictions with those estimated from the Spike protein structure. We could observe reasonable correlations ranging from $R^2 \sim 0.2$ to ~ 0.8 (see figure below) between these estimates of structural features (FoldX predictions and surface accessibility). To compare with a more naive model, we made the same predictions using the average values observed for that amino-acid type. In general we observed that the subtype based predictions (Figure below, green, subtype) tended to improve on the predictions made from the averages observed for each amino-acid (Figure below, red, wt). In particular we observed a substantial gain in the predictions for: all atom relative surface accessibility (all_atom_rel), predicted change in stability contributed by electrostatic energy, ionisation energy and side-chain h bonding energy. In summary, this tool now allows for the annotation of future DMS studies with predicted amino-acid subtypes and their biophysical properties which may be of particular use when a structural model is not available for the protein of study.

In addition to running this tool on a single new study, it is also possible to analyse groups of studies and to generate a new deep mutational landscape and list of amino-acid subtypes. This will facilitate the expansion of the repertoire of amino-acid subtypes as the number of studies increases in the near future. We hope the reviewers agree that this software package adds substantial value to this work.

2.Many of the key findings - subtypes and examples of what they might mean for specific residues of the model proteins - are difficult to parse because so much is not explained in the main text. This is true throughout the manuscript.

We agree with the reviewer that there were several aspects of the work that could be described more clearly and that there was unnecessarily reliance on technical jargon. We have tried to explain critical aspects of the analysis for those less familiar with the technical details.

a.For example, SIFT scores are introduced, and a definition of SIFT scores is given, but the interpretation of SIFT scores is not clear for readers unfamiliar with SIFT scores. A brief description of interpretation SIFT scores (such as theoretical range, what the edges of the range represent) could be beneficial.

We now describe how SIFT estimates the impact of a mutation and what the range of the scores represent.

b.Another example: it is stated that "all studies span UMAP space so the dimensionality reduction demonstrates a lack of batch effects" but it is unclear how a position would not span UMAP space if that position was used in the creation of UMAP space.

We agree this was unclear and have expanded on this to explain better what the distance in the UMAP space represents and why it is important that the positions don't separate in this space by the protein or study of origin.

c. A third, less complex example: In figure 3B, what are P and O? Permissive and Outlier? If so, state this. In Figure 3C, state that the larger graph is a Pearson correlation and the smaller heatmaps are the ER. In the text, put this in context - what does this tell the reader?

We have added the information to the figure legend and revised the corresponding text section in the results to better explain what the clustering in figure 3C represents.

3. On a related note, to point 2, this is at its core a data science paper, heavy on statistics. However, the goal (presumably) is for the researchers who experimentally mutate proteins to learn from it. As written, the heavy use of statistical terms and other field-specific jargon without additional explanation or context makes it difficult to parse by any who are not experts in those areas. There were too many examples to list but a few are here:

We agree with the reviewer that it would be important to reduce the jargon to be able to reach a wider audience.

a. Porter5 is first mentioned on page 9 without context, and then again on page 17 with a little more context, and then again in Figure S8 caption. Before its first use, there should be a brief sentence describing what Porter5 is and why it was chosen compared to any other options for secondary structure.

We now state that we obtained secondary structure predictions using Porter5. There are a very large number of secondary structure predictors and this happens to be a recently updated and well performing method that has been benchmarked against other such tools. Most approaches do, in any case, perform well at secondary structure predictions.

b. It is stated that: "strong role of mean normalised ER and its link to overall evolutionary conservation at a position (figure 2B)." There is nothing in this figure that highlights this fact to me, though I see the UMAP2 vs UMAP1 data. This needs to be more clearly labeled, and/or better explained in the text. (This goes back to the SIFT scores and their use.)

We have explained in more detail what the SIFT results are and we expanded the figure legend to make the point clearer.

c. Despite all of the technical terms, the subject characterizations of Figure 3E fall flat due to the inclusion of so many statistical calculations. What are the numbers that correlate to "most selective" or "permissive"? Include or reference the supplemental figures that show this data, and give more explanation and/or context in the main text.

We have attempted to clarify what we mean by most selective and most permissive subtypes in the text. We agree with the reviewer that it was previously not possible to quickly identify which of the subtypes from each of the amino-acids was the most selective or permissive. We have added this information to the corresponding figure legend.

d. The details on the subtypes are included in the supplemental figures, but a more detailed explanation would be helpful. For example, it would even help to supply a more succinct explanation that the 1 subtype is not the same for each amino acid and that each number subtype can have different physical meaning for an amino acid.

It is quite difficult to go into many details about the subtypes in the main text and we decided

instead to provide a few examples in the main and leave additional information in the supplementary material for those that want to delve into them. Per the reviewer's suggestion we now explain that the numbering of the subtypes refers to the subtypes of each amino-acid that occur from the highest to lowest frequency (1 .. N).

4. In the discussion, the authors refer to their work as a way to check amino acid functions across the proteome, but it's a small sample of proteins selected without relation to each other, and it is not clear that these cover the proteome's multidimensional space adequately, nor is any evidence given to this effect. If this is too broad of a claim (as I suspect), it is still useful to have it dialed back to the fact that this is the start of such a reference database. To that end, some explanation is given as to why these proteins are chosen, but how does that connect to understanding the proteome? A large number of this study's proteins are derived from *H. sapiens*, with the remainder dominated by bacteria and fungi; what of other organisms?

We certainly agree with the reviewers' concern that this is still a small sample of any proteome and we have tried already to discuss the fact that this is a starting point and that it will need to be updated as new deep mutational studies are generated. We have further expanded on this in the revised version of the discussion as per the suggestion. If the reviewer feels still that this needs improving we can further adapt it.

5. On a related note to point 4, whenever exploring fitness landscapes, the "fitness" definition is extremely important. For example, enzyme activity varies significantly across conditions, and an amino acid that is permissive under the conditions tested may not be so under an alternative (still useful) set of conditions. This was nicely acknowledged on page 6 when introducing the approach, using membrane proteins as the example, but I still struggle to see how the authors can be confident in the results without a much larger data set. Unknown nuances of this variety mean that the findings in this work are only good for explanations of proteins of the types profiled, and in the biophysical environments that were used to generate the original landscapes. This limits the utility of the findings. Note that I am not saying the results are not useful at all, but the authors may wish to make clear HOW to use the findings appropriately.

We agree with the reviewer that the selection criteria used for the experiments can have an impact on the results and it remains unclear what the extent of the impact will be. This is the case in the two examples where we could compare directly different selections for the protein such as the ones performed for HSP90 ($r^2=0.4038$) and UBI ($r^2=0.4676$). The observations we have made seem to relate to general features of amino-acid biochemistry and do not appear to be specific to any given protein. This can be seen by the fact that no single protein or proteins from a given species are segregating in a particular way in the low dimension representation of the data. Most likely, additional selection criteria will shift some positions of a given protein across different clusters but we suspect it will not strongly impact on the overall clustering itself. That is, we would still expect that some positions will not tolerate, for example charged residues, but changes in the selection pressures could shift which exact positions fit within this function. However, until larger datasets are available across multiple selection criteria, there won't be a conclusive answer. We hope the reviewer agrees that this does not detract from the impact of the key idea presented here in regards to how these datasets can be jointly analysed and how these can be expanded in the future. We think this work also very nicely underscores the importance of doing DMS experiments as we weren't able to derive similar observations simply from proxies derived from

alignments. We have added a note to the discussion to emphasize that there may be changes due to the selection criteria and that this will be an important area of future investigation.

Minor Comments

1. The common score is denoted the enrichment ratio (ER) score and represents the enrichment of a variant during selection relative to the change observed in wild type, such that positive scores are advantageous and negative ER scores are deleterious, but in nearly all the figures it appears that ER is always less than 0, except perhaps on the "Not Proline" heatmap in Figure 3. Presumably there are very few advantageous mutations compared to WT in the datasets used, but if this is the reason, it should be explicitly discussed.

The reviewer spotted correctly that the average ER scores for the subtypes are generally either neutral (tolerating) or negative (not tolerating) and only rarely positive (preferring). As per the intuition of the reviewer, the ER scores are not distributed equally towards the negative and positive site as there is a longer tail towards the negative scores as exemplified by the examples shown Figure 1C. This is in line with the biological expectation that non-neutral mutations are more likely to destabilise than stabilise or improve the function of the protein. We added this point to the discussion section.

2. "either selected the most appropriate condition" - need some reference to what was done to decide what was appropriate

We have expanded to say that we selected the conditions by favouring those that select for broader aspects of protein function instead of more specific aspects such as selection for protein interactions. For example, we used measurements of overall BRCA1 E3 ubiquitin ligase activity rather than BARD1 binding from Starita et al. (2015) as it may reflect protein function more directly.

3. Section "A unified deep mutational landscape": "The scores from these results were correlated" - r^2 less than 0.5 seem weak for correlation.

We agree that $r^2 \sim 0.4$ are not optimal for replicates but these are not strictly replicates since this includes differences in the selection approaches. We have moderated the claim to say these are sufficiently correlated and explain that the differences can be in part due to different assays and selection pressures. For example, one study measured UBI fitness via surface display and E1 ubiquitin ligase activity and the other used growth in the absence of WT UBI. For HSP90, one study used a HSP90 allele that was temperature sensitive and was rescued with the library and the other used a HSP90 that could be down-regulated and rescued with the library.

4. Figure 1B: Need x-axis for Raw graph

This has now been added.

5. Figure 2C: the 6 is cut off on the x-axis of the graph

This has now been corrected.

6. Figure 2 caption is light gray instead of black

We kept the formatting of the captions in light grey simply to make it easier to separate this

text from the main manuscript text. This will be formatted appropriately in production if the manuscript is accepted.

7. Throughout Fig 4 - 6, it was not always clear that the y-axis was a specific subtype for that amino acid. That needs to be clearer

We have tried to clarify this in the corresponding figure legends.

8. Section "Mapping the diversity of amino acid subtypes": "Permissive positions have low ER scores for all substitutions" - Why do permissive positions have low ER? It seems that all the mutations highlighted in the paper are worse than wild-type.

This was incorrect in the text. The permissive positions have low *magnitude* ER scores. We have corrected this now.

9. How does this approach handle post-translational modifications and their impact?

Post-translational modified residues were in fact one aspect where we hoped we could showcase the idea of subtypes. We tried to identify potentially known phosphorylation sites within the dataset but found only 52 such positions across different amino-acids which may not all be phosphorylated and critical for the functions probed in these assays. We did find one subtype of cysteine that is enriched for disulphide bonds which is a form of post-translational modification. Overall we think the dataset is still not large enough but that in the future the functional relevance of modified residues may also be better understood through such analysis. This would allow us to more directly compare modified residues with subtypes of other residues etc. This is discussed to some extent in the manuscript but we are happy to extend on this further if the reviewer would find this relevant.

Significance

The advance here is the combining and normalizing of mutational datasets. As noted above, however, it is not obvious to me that in so doing, new insights are pulled from the data. Toward Major comment 4, can the authors perhaps point to findings that give an insight that required the combined power of the 30 datasets and does not come out of analysis of a single protein's mutational landscape?

We fully agree with the reviewer as to what is the main advance here. We hope this study further convinces others that DMS can be jointly integrated and there will be much to gain from an expanded set of DMS studies across a larger set of proteins/conditions.

Specifically, the identification of the subtypes and importantly also their frequency across proteins is something that we think could not be achieved by the study of any single protein. We have now added in supplementary figure 7 the number of identified amino-acid subtypes obtained when subsampling the dataset. Even starting with 1000 protein positions, which is larger than any single protein in the study, we would have obtained only approximately 17 amino-acid subtypes. As we mentioned already in the discussion, we don't find and would not expect to have found, new amino-acid functions. Given the many decades of detailed structural biology studies, it is unlikely that we would find such new amino-acid functions, in particular with the current size of the dataset. However, one key aspect of the work here is that we can quantify how often different amino-acid "functions" are used across these proteins. We can say statements such as "cysteine positions seem to utilise cysteine specific properties in 65% of positions or that charged residues only rely heavily on their charge in

21-27% of positions". These types of observations are only possible with the combined analysis presented here where we can jointly derive subtypes and their frequencies. We have tried to highlight throughout the text what aspects of the work and key findings that relied on joint analysis of several DMS datasets.

In order to increase the impact of the work we have developed an analysis tool that we describe in response to the first point raised. We think this tool is also a value in itself for future work done in this field.

Nonetheless, the work is unique among the fitness landscape field, and it could be transformative to combine datasets in this way.

We appreciate the positive remark.

The audience of the work as written is probably data scientists, and this needs to change so that the findings can be employed by those who study protein evolution or engineering (the areas that I work in, as a wet-lab experimentalist). Indeed, if written for this broader audience - or, even better, by suggesting ways to design mutational landscape studies such that the results could be included in future iterations of this analysis - I think that the work could be of high impact to our understanding of protein structure and function from a biophysical point of view.

We appreciate the effort of the reviewer in guiding the manuscript to be of use to a border audience. We have tried to reduce the jargon, emphasize the added value of the work and develop a computational tool that other scientists can use to analyse any deep mutational dataset. We believe this will be a powerful addition to the field.

Reviewer 2

Evidence, reproducibility and clarity

This was a very interesting study, integrating a large number of deep mutational scanning datasets to explore the functional properties of different amino acids. It is important to emphasize that it is very much an exploratory study - there is unfortunately not a lot in terms of novel biological insight.

We agree with the reviewer that, in its essence, this is still a first exploration of the power of combining DMS datasets in this way to study, among other things, the functional properties of amino-acids and how often they are used in proteins. We think there are some biological findings that relate in particular to the frequency of usage of different functions that are novel but perhaps the major strength of the work is laying a foundation for future expansions in this area. In turn, we also think this work further shows how important DMS can be beyond the study of any single protein. As we discussed in more detail in the response to reviewer 1, we have attempted to increase the impact of this study by providing a tool that other scientists can use in the analysis of any DMS dataset(s). We have also highlighted the aspects of the work that could only be derived by the joint analysis of a large number of DMS datasets. We hope these changes provide evidence to the reviewer that there is a significant advance being made here.

I have no major concerns in regards to methodology, but do note a number of points:

- The UMAP projections presented throughout this paper are not as informative as they could be. This is mostly due to high density of overlapping opaque points, grouped in such a way that 'only last plotted is seen'. This could be improved by either using smaller points or increasing transparency. An alternative is to divide the plot into small grid squares, then colour based on the average point value in each square (on a two-colour scale). Shade by density of points.

We thank the reviewer for the useful suggestion. We have changed the UMAP projections by dividing the space into a hexagon grid structure with each hexagon representing the average property of the points found within it.

- What is the threshold for similar conditions in a DMS experiment to be comparable, and thus averaged, rather than be regarded as different conditions?

We didn't strictly use a correlation cut-off to decide to merge the experiments but instead decided to merge cases where the assay was very comparable such as selection under different drug conditions for the same drug.

- The assumption that the bottom 10% of DMS results represent null mutations is not necessarily sound, not all DMS studies look at nonsense mutations, so the results may be slightly skewed in those that do vs those that do not. I would like to see some validation of this approach.

We agree with the reviewer that this is an important assumption. We are assuming that the lowest 10% of scores represent generally deleterious variants, irrespectively of the study even when the study does not contain nonsense mutations. To look at this concern more directly we have first made use of the studies where nonsense mutations were tested. We perform an analysis where we predict the deleterious impact of mutations using SIFT and FoldX for the 10% most deleterious variants in each study. In the figure below we compared the SIFT and FoldX results when we select the lower 10% of DMS scores either keeping (with10) or excluding (without10) the nonsense mutations.

We observed that the lower 10% of DMS scores is predicted to be equally deleterious when the same study has or does not have a missense mutation tested. To confirm that this would not have an impact also on the normalization procedure we calculated the normalisation coefficient for these same datasets when we either include or exclude the missense mutations during the procedure. As shown in the figure below, the estimated coefficients are essentially unchanged.

We think these results suggest that using the lowest 10% of DMS scores as a reference for

what would be the most deleterious variants is not likely to strongly differentiate between studies that have probed or not for nonsense mutations. We have not included this in the manuscript but can if the reviewer thinks this should be included.

•Figure 2A. Hemagglutinin also has a transmembrane helix, does the observed pattern hold up here as well?

This is an interesting idea. We don't see a similar separation in the UMAP space as the other membrane proteins. However there are only 21 positions in this protein that are classified as transmembrane so it is difficult to compare this with the other proteins.

•Page 10, possible typo: "This may be because they tend to occur in positions that at least require hydrophobicity, such as the core of the protein, and therefore at least strongly hydrophobic substitutions are selected against"

I think they mean that non-hydrophobic substitutions are selected against in the core of the protein?

The reviewer is correct and we have changed this.

•Possible additional analysis: Estimate of the total number of subtypes by starting with a small number of proteins in the analysis and seeing how many more are identified with each protein added back in. The function should converge towards the estimated total number.

We thank the reviewer for this very useful and interesting idea. We have performed the suggested analysis that was added as supplementary figure 7B and 7C (reproduced below). As shown below in Figure S7B, even with around 1000 positions sampled, which would be larger than any of the single proteins studied, we can only identify around 17 amino-acid subtypes (excluding permissive and outliers). This highlights how the integration of multiple datasets is critical for this task. As we sample an increasing number of positions we can identify an increasing number of subtypes but overall we still don't see a tendency for saturation which suggests that there remain many subtypes to be discovered with increasing sample size. When we subset by different amino-acids (Figure S7C) there are some where the number of subtypes increases more slowly suggesting that for some amino-acids there may be fewer subtypes left to be discovered. However, given that the number of sampled positions is still relatively small, we don't think any estimates of the true or final number of subtypes can be discussed. We have added this new analysis to the results and discussion.

Figure S7 - A: Frequency of each subtype number fit by an exponential function, across all amino acids. $y = e^{-(0.56197 \pm 0.01655)x}$, $r^2 = 0.9359$, $p < 2.2 \times 10^{-16}$. **B:** Total number of subtypes identified when reclustering with an increasing number of positions from the dataset. The clustering algorithm was applied to increasingly large data subsets (increments of 200, starting at 1000) for 100 shuffles of the data. The mean and standard deviation of the number of functional subtypes (not permissive or outliers) identified at each position count from these samples is shown. **C:** Number of subtypes identified for alanine, cysteine, histidine and tryptophan, illustrating the saturation that occurs for some amino acids but not others.

·The authors state throughout that they are using SIFT, but we see in the methods that it is actually SIFT4G. They can be quite different in their results, so I think it would be better to call them SIFT4G scores throughout.

We have revised this throughout the text.

·What does AP stand for in Fig 3?

AP represents the alanine positions (A) that are very permissive (AP). We have tried to clarify in the figure legend that “Subtypes are numbered from most to least frequent within each amino acid, with P and O representing permissive and outlier positions”.

·"We filtered positions with scores for less than 15 of the possible 20 nonsynonymous substitutions (including nonsense) to focus on positions with sufficient data and imputed remaining missing data" How much data are you losing with this strict criteria? As many DMS studies only focus on amino acid substitutions possible from single nucleotide changes, you will lose all of these, so I am curious whether you could still get meaningful amino acid subtypes when considering such datasets.

We agree that this is an important point to clarify. We have excluded only 626 positions (or 9% of the starting set) using this filter and then imputed 1.84% of missing values for the positions that were left. We have added these numbers to the results section.

·"31 of 66 positions covered by multiple studies were assigned to the same subtype". First, this seems like a low number of positions, given that it presumably includes both HSP90 and UBI datasets, unless these have very low coverage. Second, it seems like fairly mediocre consistency - it is statistically significant, but suggests that an accurate subtype cannot be determined for more than half of positions.

We agree that the number of positions is not large but both of the studies have low coverage which causes the total number of overlapping positions to be small. While we agree that the consistency of subtypes is not high the exact selection criteria for the experiments is also not strictly the same. In addition, some of the subtypes can be also more similar than others. We investigated the cases where the positions are not assigned directly to the same subtype and for the majority of cases the assignment was to the subtype that was most similar. So we think the value of 31 out of 66 correct assignments is a fairly stringent definition of accuracy. Accounting for some degree of overlap in function between some of the subtypes would result in a higher apparent accuracy. In part, this relates to the discussion of how one decides to use this approach to define amino-acid functions as either a continuum or attempting to define clusters. We have already attempted to include in the discussion that a view of description of amino-acid functions as a continuum based on the concept of mutational landscape presented here is also appropriate and worth pursuing. We have added to the results section a small additional description of how some of the incorrect assignments occur towards the most similar subtypes.

·BLOSUM62 is an amino acid substitution matrix - I don't understand how it can provide scores for synonymous changes, so I must not be understanding how it is used here (unless you mean score of 0 for a synonymous change).

The reviewer is correct in that BLOSUM62 is an amino acid substitution matrix. However, it also has information on the probabilities that a given amino-acid is maintained (no change or synonymous change). This value within the BLOSUM62 varies across the different amino-acids and relates to how detrimental on average it would be to mutate away from that amino-acid. Similarly, while the normalised DMS scores derived here score 0 for the mutation to self it is also possible to estimate how deleterious it would be to mutate the amino-acid to any other amino-acid (average ER score for any other mutation). These two estimates are observed to be significantly correlated ($r^2=0.3191$, $p=0.009452$). We have revised this section of the results in order to make this clearer but we may also simply remove this if the reviewers feel it is complex and does not add much.

·Do synonymous changes always have $ER=0$, because differences in fitness due to synonymous differences are sometimes observed in DMS studies?

In the studies used here, the synonymous changes are almost always close to zero. 4% of synonymous have abs values > 0.1 , 92% have values < 0.05 and 59% are exactly 0. We have added these numbers of the manuscript.

Significance

I do believe that the paper is both interesting and useful, and the authors do stress several times that their results shouldn't be taken as conclusive, since addition of more data will likely increase the number of amino acid categories they identify. Given this, it seems to be presenting the clustering methodology and giving examples of uses rather than analysis of meaningful results. (This is actually in line with their stated objectives, I just thought they could have gone further.)

We appreciate that both reviewers find our work generally a useful addition to the field. As both reviewers have noted, this is a first attempt that we think of as an original piece of work that will be updated as new data is generated. We have tried to expand the discussion to highlight what aspects of work were only possible because of the integration of the different datasets. In particular we note that the frequencies of amino-acid subtypes or functions is something unique to this work. This allowed us to state to what extent protein positions rely on certain properties. As the reviewer noted we have tried to remain cautious in making broad claims exactly because we think this will be revisited as additional datasets are generated but we think this already showcases the usefulness of such an idea. We also agree that we should attempt to raise the impact of this work. We think the addition of the software tool that we have described in detail in the response to reviewer 1 is an important improvement.

Manuscript Number: MSB-2021-10305R, Exploring amino acid functions in a deep mutational landscape

Thank you again for submitting your revised study Molecular Systems Biology along with the referee reports from Review Commons. We have now heard back from the two reviewers who were asked to evaluate your revised study. As you will see below, the reviewers are satisfied with the performed revisions and are supportive of publication in Molecular Systems Biology. Reviewer #1 still lists a couple of minor concerns, which we would ask you to address in a revision.

We would also ask you to address some remaining editorial issues.

Reviewer #1:

My comments were largely addressed in this revision. I have two new minor comments:

1. First paragraph of the results section, "For example, we used measurements of overall BRCA1..." lacks context and so did not add much of an explanation for me. Perhaps add a clause, eg "For example, for studies on [protein type], we used..."
2. Second paragraph of the results section, "Three genes (HSP90, TEM1, UBI)..." These gene names correspond to proteins of a particular function (eg chaperone, beta-lactamase, and ubiquitin) and this function is needed to understand the remainder of the paragraph which discusses biological properties. It would help to add a phrase or sentence for each, describing the class of protein each gene makes, so the activity assays described (especially for UBI) make more sense to the reader.

Reviewer #2:

I am happy with the authors' responses to my original comments, and think that the addition of the new R package is likely to be useful to the community. I therefore support publication in MSB

Please find below a point by point response to the reviewers' concerns, our responses in blue.

Reviewer #1:

My comments were largely addressed in this revision. I have two new minor comments:

1. First paragraph of the results section, "For example, we used measurements of overall BRCA1..." lacks context and so did not add much of an explanation for me. Perhaps add a clause, eg "For example, for studies on [protein type], we used..."
2. Second paragraph of the results section, "Three genes (HSP90, TEM1, UBI)..." These gene names correspond to proteins of a particular function (eg chaperone, beta-lactamase, and ubiquitin) and this function is needed to understand the remainder of the paragraph which discusses biological properties. It would help to add a phrase or sentence for each, describing the class of protein each gene makes, so the activity assays described (especially for UBI) make more sense to the reader.

We have revised these two sentences as requested.

RE: MSB-2021-10305RR, Exploring amino acid functions in a deep mutational landscape

Thank you again for sending us your revised manuscript. We are now satisfied with the modifications made and I am pleased to inform you that your paper has been accepted for publication.

Corresponding Author Name:

Journal Submitted to:

Manuscript Number: